# Understanding Memory in Neural Networks through Fisher Information Diffusion

## Abstract

Information retention and transmission are fundamental to both artificial and biological neural networks. We present a general theoretical framework showing how information can be maintained on dynamically stable manifolds that evolve over time while preserving the geometry of inputs. In contrast to classical memory models such as Hopfield networks, which rely on static attractors, our approach highlights evolving stable subspaces as the substrate of memory. A central contribution of our work is the use of dynamic mean-field theory to uncover a new principle: operating at criticality (spectral radius $\approx 1$) is necessary but not sufficient for reliable information retention. Equally crucial—yet overlooked in prior studies—is the alignment between the input structure and the stable subspace. The theory leads to simple initialization rules that guarantee stable dynamics at the edge of chaos. We validate these rules in basic recurrent networks, showing that Fisher information–optimized initialization accelerates convergence and improves accuracy in sequential memory tasks, including the copy task and sequential MNIST compared to standard random initialization. Together, these results provide both principled design guidelines for recurrent networks and new theoretical insight into how information can be preserved over time.

## 1 Introduction

Recurrent neural networks (RNNs) are fundamental models for processing sequential data, and their dynamics have been a longstanding focus in both neuroscience and machine learning. Early work on random networks established that criticality and chaos play central roles in determining memory lifetime and information retention (Sompolinsky et al., 1988; White et al., 2004; Ganguli et al., 2008). Building on this foundation, subsequent approaches have sought to stabilize recurrent dynamics through architectural constraints, such as unitary and orthogonal parameterizations (Arjovsky et al., 2016; Jing et al., 2017), or through adaptive state-space models with learnable dynamics (Karuvally et al., 2025). Other theoretical directions have explored modular assemblies of RNNs (Kozachkov et al., 2023), traveling-wave dynamics as carriers of short-term memory (Keller et al., 2024), and input-driven circuit reconfiguration near criticality (Magnasco, 2025). Together, these works underscore that memory and stability emerge not from isolated units, but from the interplay between structured connectivity, dynamical regimes, and input geometry.

Traditional analyses of information capacity, such as Hopfield networks, assume that information is stored in stationary fixed points of the network dynamics. While these models have been influential, they are insufficient for modeling working memory in recurrent architectures more broadly. Unlike long-term memory, which can tolerate compression or abstraction, working memory requires preserving the fine-grained distinctions and relational geometry of inputs—maintaining not only class-level information but also the detailed differences between stimuli over short timescales. A static attractor framework would imply that memory corresponds to retrieval from a finite set of stored states, invariant across trials, which fails to capture this geometry-preserving requirement. Notably, recent neurophysiological evidence shows that working memory in the primate cortex is supported by stable dynamic manifolds with rotational dynamics (Ritter & Chadwick, 2025).

Despite these advances, the theoretical understanding of information dynamics in recurrent networks remains limited. Most analyses assume either fully dense i.i.d. connectivity or a single structured population, oversimplifying both modern architectures and biological circuits. What is missing

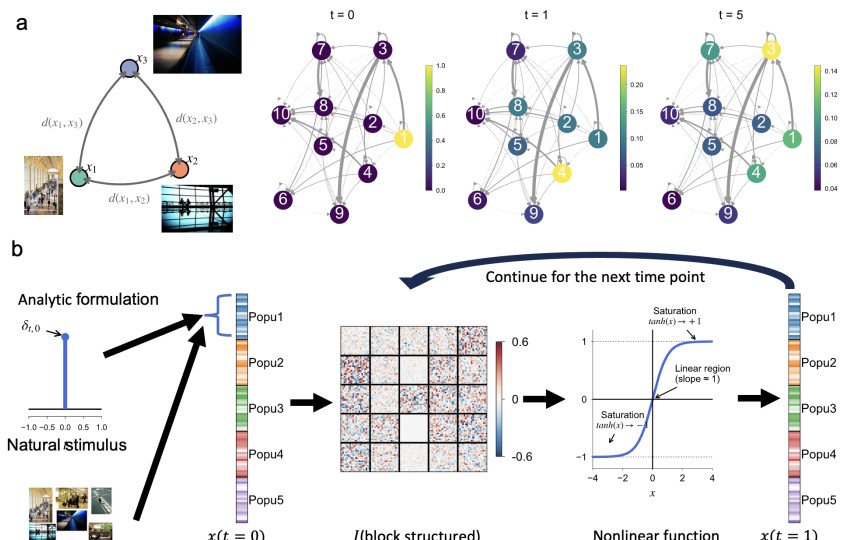

Figure 1: **Illustration of Fisher information diffusion and experimental setup.** **(a)** Unlike traditional graph diffusion models, which track the spread of neural activities, we study the diffusion of Fisher information, quantifying how well each subpopulation retains the geometry of the data (i.e., pairwise sample distances ($d(x_i, x_j)$) where $x_i$, $x_j$ are input samples) over time in a recurrent multi-subpopulation network. **(b)** Schematic of the network architecture and its time evolution. The network consists of multiple subpopulations with recurrent connections drawn from a zero-mean Gaussian distribution with specified variance, representing connection strength. Inputs can be flexibly configured; for clarity, we show the case where either an impulse (for analytic derivation) or natural images (for testing) are provided to the first subpopulation.

is a general framework for describing how information propagates and is preserved in networks composed of multiple interacting subpopulations with distinct connectivity statistics. Filling this gap is essential for understanding the mechanisms of working memory and for deriving principled design rules that optimize information retention in recurrent architectures.

**A Framework for Information Dynamics**  To this end, we introduce a framework that models general networks as block-structured systems of interacting subpopulations. Layers or modules are concatenated into a one-dimensional vector, and their interactions appear as blocks in the overall connectivity matrix (Fig. 1b). Feedforward architectures emerge as a special case with only adjacent connections, while more general configurations—including feedback loops and skip connections—are naturally encoded through the recurrent interactions among subpopulations. Within this unified representation, we analyze Fisher information dynamics directly in the space of block-structured connectivity. Drawing an analogy to thermal physics, we treat the propagation of Fisher information as a diffusion process across subpopulations (Fig. 1a). The resulting *Fisher diffusion operator* provides an analytic, Markovian characterization of how information about inputs is retained and transmitted.

In this dynamic view of the information in terms of diffusion, the theory shows that stimulus representations in recurrent networks are not fixed but evolve continuously on low-dimensional stable manifolds. While these representations change as the network unfolds in time, the geometry of the input space—defined as pairwise distinctions between stimuli—remains preserved. This dynamic perspective provides a more flexible notion of memory, where information is retained not as static states but as evolving trajectories that maintain the relational structure of inputs.

We validate the theoretical predictions in networks with block-structured connectivity matrices, where the variance of Gaussian weights in each block is controlled. We show that the Fisher diffusion operator accurately captures both the magnitude and temporal dynamics of Fisher information across subpopulations. Extending to natural image inputs, the theory also predicts how information about input geometry—pairwise distances between images—is preserved.

Finally, because representations in our framework are not fixed-point attractors, we ask whether such networks can support memory for sequences of stimuli. From our theory, we derive simple initialization rules that place the network at the edge of chaos while maximizing Fisher information retention. When tested on simple recurrent architectures, these Fisher information–optimized initializations consistently yield faster convergence and higher accuracy on sequential memory tasks—including the copy problem and sequential MNIST—compared to standard random initialization.

In summary, our framework generalizes beyond i.i.d. or single-population analyses, provides an analytic tool for studying information dynamics in recurrent networks, and offers principled initialization rules that directly connect network connectivity, criticality, and Fisher information flow.

## 2 RECURRENT NETWORKS WITH BLOCK-STRUCTURED CONNECTIVITY

We begin with the general dynamics of recurrent networks, then introduce a block-structured generalization that enables analysis at the subpopulation level.

**Recurrent dynamics.** We consider a recurrent network of $N$ neurons with discrete-time dynamics

$$h_i(t) = \sum_{j=1}^{N} J_{ij} S_j(t) + \eta_i(t), \qquad S_j(t) = \phi[w_j x(t) + h_j(t-1)], \tag{1}$$

where $h_i(t)$ is the internal state of neuron $i$, $S_i(t)$ its output, and $J_{ij}$ is the connectivity matrix connecting the neuron $j$ to neuron $i$. The activation function is $\phi(x) = \tanh(x)$ with $\phi'(0) = 1$. The network receives an external input $x(t)$ through weights $w_i$, and each neuron is driven by independent Gaussian noise $\eta_i(t)$ with zero mean and covariance $\langle \eta_i(t)\eta_j(s) \rangle = \sigma^2 \delta_{ij}\delta_{ts}$.

**Subpopulation structure.** Classical analyses typically assume that the connectivity matrix $J$ is i.i.d. Gaussian (Toyoizumi & Abbott, 2011), corresponding to a single homogeneous population. To capture more general network structures, we partition the network state $\mathbf{h}(t) \in \mathbb{R}^N$ into $M$ subpopulations (Fig. 1b). Each neuron $i$ is assigned a label $m(i) \in \{1, \ldots, M\}$, with subpopulation $m$ containing a fraction $f_m$ of the neurons such that $\sum_{m=1}^{M} f_m = 1$. The resulting connectivity matrix $J$ acquires a block structure, where each block encodes connections between two subpopulations. This formulation generalizes standard feedforward or layered networks: purely feedforward connectivity appears as a special case, while feedback and skip connections are naturally represented by off-diagonal blocks.

**Block-structured connectivity.** Within the mean-field approximation, weights are modeled as independent Gaussians with zero mean and block-dependent variances: $\langle J_{ij} \rangle_J = 0, \langle J_{ij}^2 \rangle_J = \frac{1}{N} g_{m(i)n(j)}^2$. Here $m(i)$ and $n(j)$ denote the subpopulations of neurons $i$ and $j$. Each block encodes connections from subpopulation $n$ to $m$, with variance parameter $g_{mn}^2$ controlling its strength. We will refer to the block-gain matrix $G$ with entries $G_{mn} \equiv g_{mn}^2 f_n$.

**Fisher information.** We probe memory with an impulse input $x(t) = \theta \, \delta_{t,0}$ and study how well its amplitude $\theta$ is preserved across time and populations. The Fisher information (FI) about $\theta$ is

$$\mathcal{I}(\theta, t) = \mathbb{E}_{p(\mathbf{h}(t)|\theta)} \left[ \frac{\partial^2}{\partial \theta^2} \log p(\mathbf{h}(t) \mid \theta) \right], \tag{2}$$

which defines the Fisher memory curve (Ganguli et al., 2008).

For a fixed $J$, $p(\mathbf{h}|\theta)$ is Gaussian, the FI simplifies to (see Appendix A.2.6):

$$\mathcal{I}(\theta, t) = \left\langle \frac{\partial}{\partial \theta} \frac{\partial}{\partial \theta} \log p(\mathbf{h}(t)|\theta) \right\rangle_{p(\mathbf{h}(t)|\theta)}, \tag{3}$$

Mean-field theory yields a block-diagonal covariance $\left\langle \Sigma_{ij}(t) \right\rangle_J = \delta_{ij}(q_{m(j)})$ with population-specific variances $q_{m(j)}$, giving (Appendix A.2.6)

$$\mathcal{I}(\theta, t) = N \sum_m \frac{f_m}{q_m} \left\langle \left( \frac{\partial \mu_m(t)}{\partial \theta} \right)^2 \right\rangle_J, \qquad q_m = \sigma^2 + \sum_n G_{mn} \langle S_n^2 \rangle_n, \tag{4}$$

where $\mu_m(t) \equiv \langle h_m(t) \mid J \rangle$. Although $\mu_m(t) = 0$ on average, FI remains nonzero since it depends on the variance of the sensitivity $\partial \mu_m(t)/\partial \theta$, which is shaped by both the nonlinearity $\phi$ and the inter-population connectivity. Importantly, this sensitivity is not constant over time and differs across populations.

**Fisher information diffusion.** We now summarize the key result (derivation in Appendix A.2). We have derived the Fisher diffusion operator $A$ that propagates sensitivities $\langle (\partial \mu_m(t)/\partial \theta)^2 \rangle_J$ across subpopulations from one time step to the next. For two subpopulations, it factorizes into a connectivity and a sensitivity term:

$$A = \underbrace{\begin{pmatrix} G_{11} & G_{12} \\ G_{21} & G_{22} \end{pmatrix}}_{\text{Connectivity}} \cdot \underbrace{\begin{pmatrix} \langle (S')^2 \rangle_1 & 0 \\ 0 & \langle (S')^2 \rangle_2 \end{pmatrix}}_{\text{Sensitivity}}, \tag{5}$$

where $\langle (S')^2 \rangle_n$ denotes the mean squared derivative of the activation function in subpopulation $n$. The connectivity block captures how information is routed between groups, while the sensitivity block captures how nonlinear responses modulate this transfer. In Appendix A.2 we provide an analytic expression for $\langle (S')^2 \rangle_n$ that depends solely on the block-gain matrix $G$, making the operator fully determined by network structure. In the linear limit—when the activation function is purely linear—one has $\langle (S')^2 \rangle_n = 1$ for all subpopulations. In this case the sensitivity block reduces to the identity, and the Fisher information diffusion operator $A$ coincides with the block-gain matrix $G$ itself. Repeated application of $A$ describes how information flows across populations. The total FI at time $t$ is then $\frac{\mathcal{I}(t)}{N} = \sum_{mn} \frac{f_m}{q_m} (A^t)_{mn} w_n^2$, and $\mathcal{I}(0) = 0$.

Crucially, although individual neural activities evolve nonlinearly, the collective statistics of subpopulations can be expressed as evolving linearly under the diffusion operator. The nonlinearity is absorbed into the term $\langle (S')^2 \rangle_n$, which itself is a nonlinear function of the block gains $G$. In this way, the operator acts analogously to a transfer matrix in graph diffusion, providing a linear structure that enables a tractable analytic description of how connectivity shapes the encoding and preservation of information over time. We present two tests to evaluate our analytic characterization of Fisher information.

**Direct Fisher quantification.** First, we directly estimate Fisher information from network simulations. We simulate a network with $N = 10,000$ neurons, $f_1 = f_2 = 0.5$, $\sigma = 0.1$, and input weights $w_1 = 1, w_2 = 0$, so that the impulse is applied only to the first subpopulation. For input amplitudes $\theta \in \{0, -0.1, 0.1\}$, we compute the sensitivity term $\langle (\partial \mu_m(t)/\partial \theta)^2 \rangle_J$ in Eq. equation 4 (see Appendix A.3). To test how connectivity motifs affect information dynamics, we consider three configurations: (i) self-recurrence only, (ii) feedforward coupling, and (iii) feedback coupling (Fig. 2a–c). Across all cases, the analytic solution based on the diffusion operator accurately matches simulations—capturing not only the magnitude of Fisher information in each population, but also the temporal dynamics, including oscillatory flow between subpopulations. Finally, we examine how the agreement scales with network size $N$: the mean-squared error between simulated and analytic trajectories decreases rapidly and becomes negligible for $N \geq 1000$ (Fig. S3).

**Preservation of input geometry.** We next tested whether Fisher information predicts how well a network preserves the geometry of natural inputs—equivalently, the pairwise distances between input stimuli. Below, we provide an intuitive argument showing that Fisher-optimal connectivity conditions coincide with those required for local isometry preservation.

**Theorem 1** (Connecting Fisher information with preservation of geometry). *Consider a recurrent network with block-gain matrix $G$ and activation function $\phi$. Under the mean-field approximation, optimal information retention—defined as preservation of local geometry between stimulus representations—is achieved when $G \langle \phi'^2 \rangle = 1$. This is precisely the Fisher-information criterion for non-vanishing memory.*

*Sketch proof.* The result follows by connecting ideas from compressed sensing and nonlinear mean-field theory.

*(1) Linear case and RIP.* For a linear map $f(x) = Jx$, the Restricted Isometry Property (RIP) ensures approximate distance preservation: $\|f(u) - f(v)\|^2 \approx \|u - v\|^2$. For Gaussian weights $J_{ij} \sim \mathcal{N}(0, g^2/N)$, RIP holds when $J^\top J \approx I$, which requires $g^2 = 1$.

*(2) Nonlinear extension.* With nonlinearity $\phi$, local distances transform as $\|f(x) - f(x')\|^2 \approx \|\phi'(x) J(x - x')\|^2$. Replacing $\phi'(x)^2$ by its population average under mean-field theory gives the effective gain condition $G \langle \phi'^2 \rangle \approx 1$.

*(3) Fisher information connection.* From the Fisher diffusion framework, sustained (non-decaying) memory requires that the leading eigenvalue of $G\langle \phi'^2 \rangle$ equals 1. Thus, the conditions for (i) preserving local geometry (via RIP/Johnson–Lindenstrauss (Foucart & Rauhut, 2013)) and (ii) sustaining Fisher information are identical. □

**Empirical validation on real images.** We presented 15,619 IndoorCVPR_09 images (flattened to dimension 7500) as inputs to the first subpopulation of a network with $N = 15,000$, $f_1 = f_2 = 0.5$, and $\sigma = 0.1$. Each image was processed individually, and at time $t$ we recorded the activity vectors of both subpopulations as the network's internal representations. To quantify geometry preservation, we computed all pairwise Euclidean distances between the original images, and likewise all pairwise distances between their corresponding neural representations (see Appendix A.4). We then measured the correlation between these two distance matrices: a correlation of 1 would indicate perfect isometry (geometry preserved exactly), while lower correlations reflect increasing distortion. This correlation therefore serves as a direct measure of how faithfully the network preserves the relational structure of its inputs (Fig. 2d–f). Although this metric is distinct from Fisher information, it produces the same qualitative conclusions: the analytic framework accurately predicts both the oscillatory dynamics of information flow and the relative ability of different network motifs to preserve input geometry.

This empirical test highlights a key difference from Hopfield networks. In Hopfield models, memory is implemented by fixed-point attractors, and capacity is limited by the number of such stable states that can be stored. In our framework, by contrast, memory is defined by how well the differences between stimuli are preserved as activity evolves. This capacity does not depend on the number of stimuli presented, but instead on whether the network size $N$ is sufficiently large relative to the sparsity of the input space—a condition closely analogous to the Restricted Isometry Property (RIP) for Gaussian matrices, which links the number of measurements to input sparsity (Foucart & Rauhut, 2013).

**Dataset-independent encoding dynamics.** A central theoretical prediction is that Fisher-optimal initialization preserves pairwise distances independently of the specific input ensemble, provided the input sparsity is below the network's effective dimensionality. To test this, we repeated the analysis using CIFAR-10 images (flattened to 7500 dimensions) (Fig. S4). The resulting dynamics closely match those from IndoorCVPR_09. The Pearson correlations between the two datasets' information-flow trajectories are extremely high: $0.993, 0.992, 0.980$ (all $p \ll 10^{-60}$), confirming that Fisher-optimal initialization yields dataset-independent encoding behavior.

## 3 CONDITIONS FOR OPTIMAL FISHER INFORMATION

In the context of information diffusion, achieving maximal long-term retention of an input requires two conditions on the diffusion operator $A$:

1. **Criticality.** The spectral radius of $A$ must satisfy $\rho(A) = \max_i |\lambda_i| = 1$. If $\rho(A) < 1$, Fisher information decays exponentially; if $\rho(A) > 1$, it diverges uncontrollably. Criticality therefore guarantees the dynamic stability Kadmon & Sompolinsky (2015) such that information does not vanish at long times, but on its own it is not sufficient for optimal retention.

2. **Eigenvector alignment and transient information.** Let $v$ denote the normalized right eigenvector associated with the leading eigenvalue $\lambda_{\max} = 1$. Asymptotically, $\lim_{t \to \infty} \mathcal{I}(t) \propto \|(\mathbf{w}^\top v)v\|_1$, where $\mathbf{w}$ is the input configuration. Thus, only the input component aligned with $v$ is retained indefinitely. From the perspective of network design, however, transient information carried by other modes should not decay too quickly. A complication is that $A$ is generally nonnormal (not symmetric), so its eigenvalues and eigenvectors may be complex. Since the input configuration $\mathbf{w}$ is real, the effective alignment with complex eigenvectors can be small, limiting long-term retention. In practice, explicitly computing eigenvalues and eigenvectors is costly for networks with many subpopulations. By contrast, computing Fisher information over time only

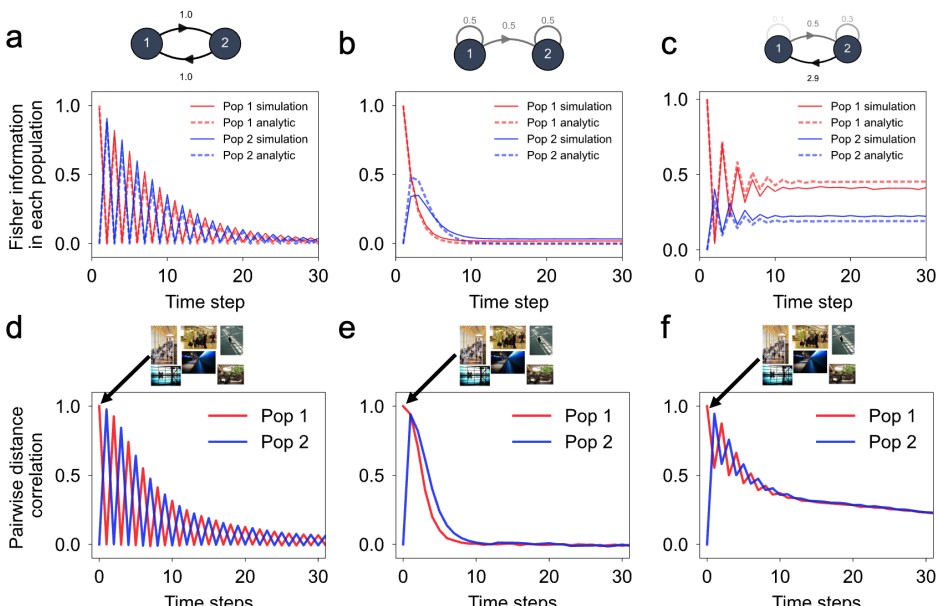

Figure 2: **Empirical validation of the Fisher diffusion framework. (a–c)** Time evolution of Fisher information in two subpopulations after an input pulse to Population 1 at $t = 0$, under three motifs: (a) self-recurrence only, (b) feedforward only, and (c) a recurrent architecture close to the analytic optimum. Solid lines show simulation results; dashed lines show analytic predictions from the FI diffusion operator. The analytic framework accurately captures both the magnitude and temporal dynamics, including oscillatory information flow. **(d–f)** Geometry preservation for the same networks for IndoorCVPR_09 images. Pairwise correlations between input distances and network representations quantify how well each architecture maintains input geometry. The results confirm that architectures predicted to optimize Fisher retention also preserve stimulus geometry more effectively.

requires iterated multiplication by $A$, which is more scalable. Moreover, by choosing the integration horizon $T$, one can tune the emphasis between transient retention and long-term stability. A practical design objective is therefore to maximize the average Fisher information over a finite horizon: $\overline{\mathcal{I}} = \frac{1}{T} \sum_{t=1}^{T} \mathcal{I}(t)$. This metric balances stability at criticality with the preservation of transient information.

To illustrate these conditions, we analyze a simple two-population recurrent network. The block gain matrix is $G = \begin{pmatrix} G_{11} & G_{12} \\ G_{21} & G_{22} \end{pmatrix}$, so that the Fisher diffusion operator $A$ depends on the four parameters $G_{mn}$. Because $A$ can be summarized in terms of its trace and determinant, the parameter space can be reduced from four to two dimensions, enabling a clear visualization. We perform a dense grid search over all $G_{mn} \in [0, 3]$. For each parameter setting, we compute: The spectral radius $\rho(A)$, used to identify the critical boundary $\rho(A) = 1$ (Fig. 3a), and the average Fisher information $\overline{\mathcal{I}}$ across 100 timesteps, aggregated over both populations (Fig. 3b). We observe that:

1. The critical boundary (grey dashed line) extends across the full range of $\mathrm{Tr}(G)$, confirming that criticality is a necessary condition for sustained information flow.

2. Optimal information retention occurs only along the critical boundary but is restricted to a narrower band of $\mathrm{Tr}(G)$, showing that criticality alone is not sufficient.

This demonstrates that while criticality is required for sustained information flow, alignment of the stable diffusion direction $v$ with the input configuration $\mathbf{w}$ is additionally necessary to achieve optimal Fisher information retention.

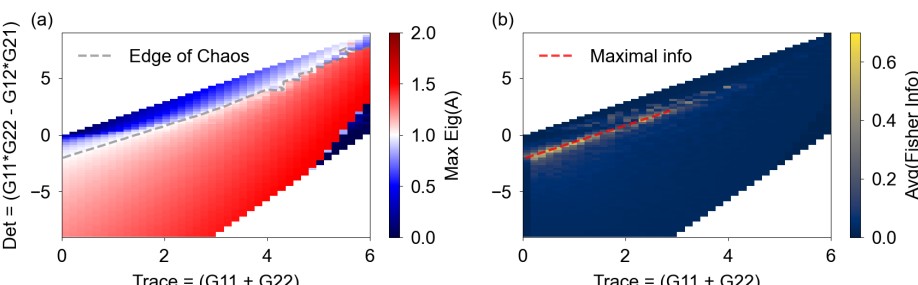

Figure 3: **Edge of chaos: necessary but not sufficient for optimal Fisher information.** (a) Phase diagram of the spectral radius $\rho(A)$ across network parameters $\text{Tr}(G)$ and $\det(G)$. The grey dashed line marks the critical boundary $\rho(A) = 1$. (b) Average Fisher information per timestep over 100 steps, showing that optimal retention occurs only within a restricted band of $\text{Tr}(G)$ values.

## 4 OPTIMAL STRUCTURE FOR LONGER CHAINS

Our analytical framework extends naturally to networks of many subpopulations or deeper recurrent structures. For clarity and tractability, we focus on sequential chains in which only adjacent subpopulations are connected on through adjustable self-recurrent, feedforward, and feedback links (Fig. 4). The connectivity is captured by a generalized Toeplitz-like gain matrix $G$ where only $G_{mm}$, $G_{m,m+1}$, and $G_{m+1,m}$ are nonzero, preserving the chain structure while allowing parameter flexibility. All subpopulations are equal in size ($f_m = 1/M$), and input is applied only to the first subpopulation ($w_i = \delta_{i,1}$).

Within this architecture we build the Fisher information functional in terms of the allowed block gains $G_{mn}$ and maximize the time- and population-averaged Fisher information $\overline{\mathcal{I}}$ over $T = 100$ using differential evolution. The optimization reveals clear design principles: strong feedforward connections propagate signals efficiently, while carefully placed feedback stabilizes and modulates this flow. Indiscriminate feedback is detrimental; instead, optimal networks exhibit sparse, strategically positioned feedback links that break the chain into nested loops for robust information retention. The characteristic broken-feedback pattern can be intuitively justified in the linear limit (Appendix A.5).

Finally, we find a striking scaling law (Fig. 4): keeping the total number of neurons fixed—yet large enough for mean-field theory—the network's total Fisher information grows approximately linearly with the number of subpopulations. Thus, deeper or more finely partitioned chains intrinsically possess greater information capacity when their connectivity is properly optimized.

## 5 SEQUENTIAL STIMULUS

For an optimal network that satisfies the necessary condition of dynamic stability, the neural activity evolves on a stable manifold rather than settling into a fixed-point attractor. When an input is projected into the network, the internal representation changes continuously over time instead of remaining static. As a result, identical stimuli injected at different times can lead to distinct downstream representations at later time. This continual drift of stimulus representations provides a natural mechanism for encoding the order of sequential inputs, akin to the function of working memory.

Once a block-gain matrix $G$ optimal for Fisher information is found for a given input configuration, a corresponding block-structured connectivity matrix $J$ can be readily constructed. Specifically, we sample the elements of $J$ from zero-mean Gaussians with variances derived from the corresponding entries of $G$. This simple procedure initializes a recurrent network to operate near the edge of chaos, providing a principled starting point for training.

To test the principles of our initialization and its relevance for sequential stimulus processing, we tested the method on two benchmarks on simple RNNs: the copy task and sequential MNIST. Our

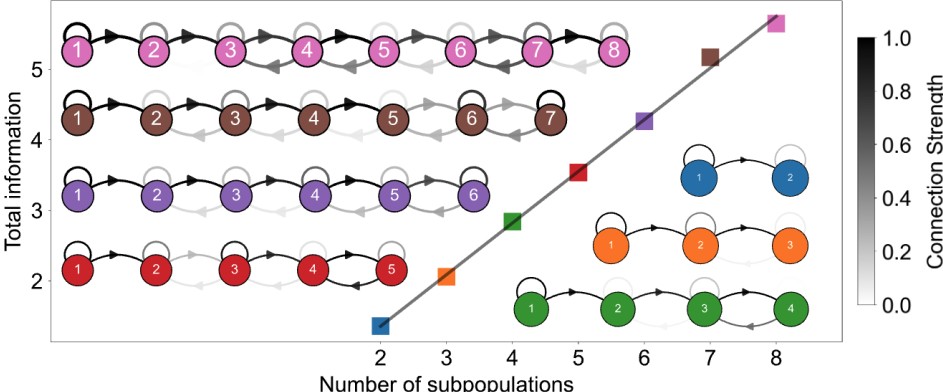

Figure 4: **Connectivity structures and information retention across subpopulations.** Inset diagrams show optimized connectivity patterns for networks with 2–8 subpopulations, revealing strong feedforward pathways, moderate self-recurrence, and sparse feedback loops. The main panel demonstrates that total Fisher information retention scales linearly with the number of subpopulations, indicating that greater modular depth enhances memory capacity.

hypothesis is that initializing a network to operate near the edge of chaos facilitates more efficient training.

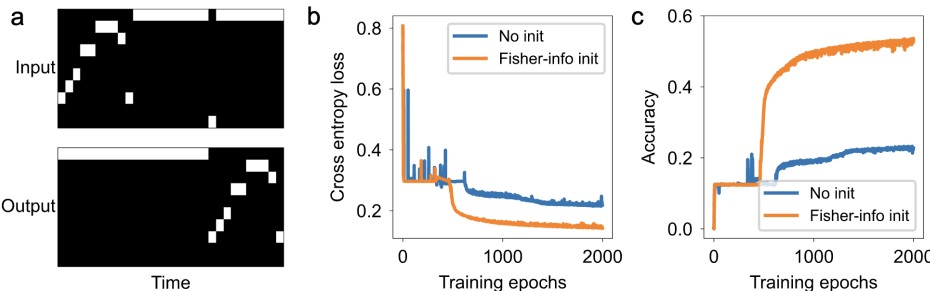

Figure 5: **Sequential memory test on the copy task.** (a) Illustration of the copy task with $T_{\text{delay}} = 10$. (b) Networks initialized with Fisher information–optimized weights converge significantly faster and achieve higher accuracy than those with standard random initialization.

**Copy task** The copy task follows the standard setup of Graves et al. (2014), Arjovsky et al. (2016), and Gu et al. (2021). Each input sequence has length $T_{\text{delay}} + 20$. The first 10 tokens are random one-hot vectors in categories $\{1, \ldots, 8\}$, followed by $T_{\text{delay}}$ zeros, a single delimiter token (category 9), and finally 9 more zeros. The target output has the same length but remains zero until the final 10 steps, where it reproduces the initial random sequence (Fig. 5). This task probes a network's ability to encode categorical information and maintain it in memory for $T_{\text{delay}}$ time steps before recall.

We trained a simple RNN of 100 neurons with a $\tanh$ nonlinearity and $T_{\text{delay}} = 50$. For Fisher-information-optimized initialization, the network is partitioned into 10 equal subpopulations (matching the input dimension), with purely feedforward connections $G_{m+1,m} = 1$ ($m = 1, \ldots, 9$) and a single feedback link $G_{1,10} = 1$. The corresponding Fisher diffusion matrix $A$ thus has spectral radius one. Weights from input to RNN are sent such that input stimulus are passed to the first subpopulation. Compared to standard random initialization, the Fisher-optimized initialization yields substantially faster convergence and higher final accuracy under otherwise identical training conditions (learning rate $= 10^{-3}$; Fig. 5).

We additionally evaluate Fisher–information–optimal initialization against widely used schemes—including Xavier, Kaiming, orthogonal, and unitary initialization—on the copy task across multiple random seeds and various delay lengths (Figs. S6, S7, S8). Across all condi-

tions, Fisher–optimal initialization consistently outperforms these standard methods, confirming the theoretical prediction that networks initialized in the Fisher–information–optimal regime preserve sensitivity to past inputs. In contrast, standard initializations are not information–optimized and require optimization to search a high-dimensional, non-convex landscape, often trapping training in suboptimal solutions.

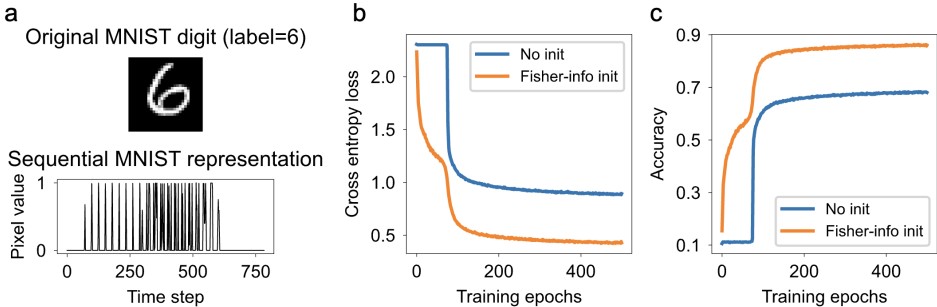

Figure 6: **Sequential memory on Sequential MNIST.** (a) Pixel values of each MNIST image are presented as a 784-step input sequence. (b) Networks initialized with Fisher information–optimized weights converge substantially faster, and (c) achieve higher final accuracy compared to networks with standard random initialization.

**Sequential MNIST**   In the sequential MNIST task, the 784 pixels of each image are fed one by one into the RNN, and the network must classify the digit using only the final hidden state (Fig. 6). To emphasize the advantage of operating at the edge of chaos, we fixed the recurrent weights to the same circular subpopulation structure used in the copy task and trained only the readout layer. In practice, using a one-layer MLP with $\tanh$ nonlinearity as the readout significantly improved performance. As with the copy task, Fisher-information-optimized initialization led to both faster convergence and higher final accuracy than random initialization (Fig. 6).

Because our initialization is fully analytic and depends only on connectivity parameters—rather than being learned from data—the theory predicts that Fisher-optimal initialization should preserve input perturbations for general sparse input stimuli. We test this by extending the sequential-classification experiments to CIFAR-10 and IndoorCVPR_09 (Figs. S11, S12; details in Appendix A.6.3). Across both datasets, and relative to Xavier, Kaiming, orthogonal, and unitary initializations, Fisher–optimal networks consistently train faster and reach higher accuracy—even with the recurrent weights fixed. This demonstrates that information preservation is governed by the connectivity structure rather than dataset-specific features, and further supports the prediction that Fisher–optimal initialization maintains sensitivity to past inputs.

Together, these findings show that initializing the network with Fisher-information–optimized weights—i.e., operating at the edge of chaos—naturally creates a stable manifold along which input representations can evolve. This initialization effectively equips the recurrent network with an intrinsic encoder that both preserves the geometry of the input stimulus and supports flexible movement of representations. As a result, training can focus on learning an appropriate decoder, leading to faster convergence and higher final accuracy.

## 6   DISCUSSION

Our contributions are threefold: (i) We introduced a block-structured, mean-field framework in which the Fisher diffusion operator analytically tracks how information flows across interacting subpopulations in recurrent networks. (ii) Criticality (spectral radius $\approx 1$) is necessary but not sufficient for long-term retention, and alignment between input structure and the stable subspace is equally essential. (iii) From these principles we derived simple, Fisher-information–optimized initializations that (empirically) accelerate training and improve accuracy on sequential memory tasks.

We bridge multiple perspectives—Fisher information, geometry preservation, and dynamical stability—under a single operator formulation. This connection explains why preserving local geometry, maintaining stability at criticality, and ensuring Fisher information flow are mathematically equivalent conditions. Importantly, the block-structured formulation extends classical one-population mean-field theory to arbitrary modular architectures, making it possible to study realistic networks with feedback, skip connections, and heterogeneous subpopulations.

The analytic expression linking connectivity structure to Fisher-information optimality yields a principled, theory-driven initialization rule for recurrent networks. A key advantage of this framework is that a large recurrent weight matrix can be configured correctly by adjusting only a small set of population-level block gains $g_{ij}$, each determining the variance of the Gaussian weights within a connectivity block. This mapping from a few block-level parameters to the full recurrent matrix places the network directly in the Fisher information optimal regime, where sensitivity to perturbations of past inputs is preserved over long timescales. Such initialization is particularly beneficial for working-memory and other short-term sequential tasks, in which late-time activity must remain sensitive to inputs presented many steps earlier. Fisher information quantifies exactly this notion of sensitivity: classical fixed-point models such as Hopfield networks intentionally collapse small perturbations to enforce convergence and therefore cannot maintain the fine-grained distinctions required for dynamic memory. In contrast, standard initialization schemes are not memory-aware and place the network in a generic region of the high-dimensional parameter space, forcing optimization to discover configurations that preserve long-range sensitivity—a process that is unstable and prone to regions where gradients vanish or explode. By initializing connectivity directly in the Fisher-optimal regime, our framework avoids these difficulties and provides a theoretically grounded method for stabilizing information flow in recurrent architectures from the outset.

Our analysis focuses on single-state recurrent networks, where the Fisher–information dynamics can be characterized analytically. LSTMs and GRUs include coupled hidden states and multiplicative gates, placing them outside the theoretical regime considered here. Nevertheless, because each gate contains a recurrent transformation structurally similar to the dynamics we analyze, we evaluated whether Fisher–optimal initialization might still offer practical benefits at the gate level. As discussed in Appendix A.6.2, we found that applying Fisher–optimal initialization to these recurrent components can improve training performance on long-range sequential memory tasks.

The dynamic, geometry-preserving memory described by our framework provides a principled alternative to classical attractor models. Instead of storing fixed points, recurrent networks maintain evolving trajectories that conserve the relative geometry of inputs—consistent with recent neurophysiological observations of stable manifolds and rotational dynamics in cortex (Ritter & Chadwick, 2025). Operating at the subpopulation level makes it naturally suited to multi-area circuits, offering predictions for how inter-areal connectivity supports information retention and traveling-wave–like activity patterns. Relatedly, recent theoretical work shows that hidden traveling waves in trained RNNs can bind working memory variables to wave-like representations (Karuvally et al., 2024), suggesting that our framework for information dynamics can also provide a foundation for understanding wave-based mechanisms of memory.

**Limitations** While our framework provides a principled and interpretable theory of information dynamics in recurrent networks, several limitations remain. Most importantly, our analysis is focused on the encoding of information: how network connectivity structures shape the retention and propagation of Fisher information across subpopulations. We do not address the decoding stage, where task-specific outputs are read out from the evolving internal representations. Thus, our framework should not be viewed as a method for directly discovering architectures that maximize task performance, but rather as a way to endow a given architecture with theoretically grounded initializations that improve training efficiency and stability. In this sense, our work is complementary to performance-oriented models such as unitary RNNs, orthogonal networks, and structured state-space models, which achieve superior accuracy on demanding sequence benchmarks. Our goal has not been to compete with such models, but to provide a general theoretical foundation that explains how information is preserved, and to derive simple initialization rules that translate this theory into practice. By doing so, we highlight principles—criticality, alignment, and balanced Fisher flow—that may also inform the design of future high-performance architectures.

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

# A APPENDIX

## A.1 RELATED WORK

### DISTRIBUTED POPULATION DYNAMICS AND WORKING-MEMORY CODES

There is now a rich literature showing that memory is implemented by distributed, population-level dynamics rather than single "memory cells." For example, Cavanagh et al. (2018); Spaak et al. (2017) analyze how working-memory representations in primate prefrontal cortex can be supported by combinations of persistent and dynamic population codes. More precisely, Spaak et al. (2017) show that during a memory-guided saccade task, despite dynamic population coding, the representational geometry of working memory remains stable. Meyers et al. (2008) demonstrate dynamic population coding of category information in ITC and PFC during delay tasks, emphasizing time-varying trajectories rather than static tuning. Finally, Kurtkaya et al. (2025) and other recent RNN studies characterize dynamical "phases" (such as limit cycles, slow manifolds, sequential activity) that support short-term memory in trained recurrent networks.

Our contribution is complementary to this line of work. The biological studies above collectively demonstrate that neural populations encode task-relevant information in the geometry of their activity trajectories, and that this information can persist even when individual neurons exhibit only transient or heterogeneous responses. Building on this assumption, we develop a closed-form theoretical framework that *quantifies* how much information about input differences can be preserved by such population dynamics. Rather than analyzing decoding accuracy from recorded neural activity, we derive Fisher-information–based expressions that make explicit how information retention depends on population-level connectivity motifs—including the trace and determinant of the connectivity matrix. In other words, whereas prior biological and computational work establishes *that* distributed neural trajectories carry short-term and working-memory information, our framework provides an analytic characterization of *how much* of the input geometry can be preserved, and *under which dynamical regimes*, directly from the parameters governing the network's connectivity structure.

### FISHER INFORMATION IN MACHINE LEARNING AND DYNAMICAL NETWORKS

Prior work has used Fisher information (FI) to analyze memory and information retention in recurrent neural dynamics, though almost entirely in homogeneous or unstructured settings. Ganguli et al. (2008) and later Ganguli & Sompolinsky (2010) applied FI to characterize how recurrent linear systems maintain short-term memory of past inputs and established fundamental limits on memory capacity. A parallel body of work in computational neuroscience has focused on linear Fisher information, which provides a closed-form expression for stimulus discriminability in recurrently connected populations (Beck et al., 2011) and has been widely used experimentally to estimate information from correlated population activity (Kanitscheider et al., 2015).

Beyond recurrent dynamics, Fisher information has a long history in machine learning as a metric that shapes optimization, curvature, and gradient propagation. Amari et al. (2019) analyses natural-gradient learning in random deep networks; Pennington & Worah (2018) characterize the eigenvalue spectrum of the Fisher Information Matrix (FIM) in wide networks; Karakida et al. (2019; 2020) study universal statistics and pathological curvature regimes induced by extreme FIM anisotropy; and Karakida & Osawa (2020) show that approximate natural-gradient methods still inherit fast-convergence guarantees in wide limits. Hayase & Karakida (2021) further demonstrate that networks satisfying dynamical isometry nevertheless develop concentrated FIM spectra that require depth-dependent learning-rate scaling, directly linking FI to vanishing and exploding gradients.

Relative to these lines of work, our contribution is different in scope and objective. Prior studies either focused on homogeneous single-population RNNs, linearized dynamics, or the role of Fisher information in optimization geometry. By contrast, our framework derives explicit, analytic conditions under which *multi-population* recurrent networks preserve Fisher information about early inputs over time, thereby maintaining the geometry of input differences along dynamically evolving trajectories.

Crucially, these conditions provide a direct link between the *connectivity structure* of the network—specifically, population-level parameters such as the trace and determinant of the $G$ matrix—and the

amount of Fisher information that can persist along recurrent dynamics. This connection has practical consequences for sequential-memory learning. Instead of relying on gradient descent to discover a narrow region of parameter space that supports long-range sensitivity to inputs, our Fisher-optimal initialization places the network in a regime that already retains information about distant past stimuli. Because large RNNs have high-dimensional, non-convex loss landscapes, standard initializations that are not memory-optimized often struggle to reach these regions and can become trapped in poor local minima where gradients vanish. In contrast, Fisher-optimal initialization tunes the block-structured gains so that a large number of individual recurrent weights are automatically set to a configuration that preserves input geometry from the outset, thereby accelerating training and improving final performance.

## A.2 ANALYTIC FISHER INFORMATION FOR MULTIPLE SUB-POPULATIONS

We consider a recurrent network divided into multiple subpopulations. For each sub-population $m$, let

$$\mu_m(t) \;=\; \left\langle h_m(t) \,\middle|\, J \right\rangle . \tag{6}$$

denote the mean activity at time $t$, averaged over dynamic noise but at fixed synaptic matrix $J$. Our goal is to compute the Fisher information $\mathcal{I}_m(t)$ of $\mu_m(t)$ with respect to an input parameter $\theta$:

$$\mathcal{I}(\theta, t) = K \sum_m \frac{f_m}{q_m} \left\langle \left( \frac{\partial \mu_m(t)}{\partial \theta} \right)^2 \right\rangle_J . \tag{7}$$

The key is to derive the analytic formula for $\left\langle \left( \frac{\partial \mu_m(t)}{\partial \theta} \right)^2 \right\rangle_J$.

### A.2.1 REPLICA TRICK FOR THE DERIVATIVE OF THE MEAN

Introducing replicas $a, b$, we have

$$\left\langle \left( \frac{\partial \mu_m(t)}{\partial \theta} \right)^2 \right\rangle_J = \left\langle \left[ \frac{\partial}{\partial \theta} \langle h_m(t)|J\rangle \right]^2 \right\rangle_J = \left\langle \frac{\partial}{\partial \theta^a} \langle h_m^a(t)|J\rangle \frac{\partial}{\partial \theta^b} \langle h_m^b(t)|J\rangle \right\rangle_J$$

$$= \frac{\partial^2}{\partial \theta^a \partial \theta^b} \langle h_m^a(t) h_m^b(t) \rangle = \frac{\partial^2}{\partial \theta^a \partial \theta^b} q_m^{ab}(t) , \tag{8}$$

where $q_m^{ab}(t) \equiv \langle h_m^a(t)\, h_m^b(t) \rangle_m$.

### A.2.2 MEAN-FIELD EXPRESSION FOR THE CORRELATION

In mean-field, the correlation splits into an i.i.d. noise term and a term generated by recurrent inputs:

$$q_m^{ab}(t, s) = \sigma^2 \delta_{ab} \delta_{ts} + \sum_n g_{mn}^2 f_n \langle S_n^a(t) S_n^b(s) \rangle_n = \sigma^2 + \sum_n G_{mn} C_n^{ab}(t, s) , \tag{9}$$

with $G_{mn} \equiv g_{mn}^2 f_n$ and $C_n^{ab}(t, s) = \langle S_n^a(t) S_n^b(s) \rangle$, the firing rate correlation in sub-population $n$.

### A.2.3 DIFFERENTIATING FIRING RATE CORRELATION FUNCTION

Applying $\partial_\theta^a \partial_\theta^b$ to $C_n^{ab}(t, s)$ and using the chain rule for the nonlinearity $\phi(\cdot)$ (with $\phi' = \frac{d\phi}{dz}$) yields

$$\partial^{ab} C_n^{ab}(t) = \partial^{ab} \langle \phi^a(t) \phi^b(t) \rangle_n$$

$$= \partial^{ab} \langle \phi(w_n \Theta^a(t-1) + x^a(t-1)) \phi(w_n \Theta^b(t-1) + x^b(t-1)) \rangle_n$$

$$= \langle \phi'^a \cdot (w_n + \partial_a x^a(t-1)) \, \phi'^b \cdot (w_n + \partial_b x^b(t-1)) \rangle_n$$

$$= \langle \phi'^a \phi'^b \rangle_n \langle (w_n^2 + (\partial_a x^a(t-1) + \partial_b x^b(t-1)) w_n + \partial_a x^a(t-1) \partial_b x^b(t-1) \rangle_n$$

$$= \langle \phi'^a \phi'^b \rangle_n (w_n^2 + \partial_a \partial_b \langle x^a(t-1) x^b(t-1) \rangle_n$$

$$= \langle \phi'^a \phi'^b \rangle_n (w_n^2 + \partial_a \partial_b \, q_n^{ab}(t-1)) . \tag{10}$$

The first order in line 4 with terms $\langle \partial_a x^a \rangle_n = \partial_a \langle x^a \rangle_n = \partial_a 0 = 0$.

### A.2.4 RECURRENCE FOR THE SECOND DERIVATIVE OF POPULATION SPECIFIC VARIANCE

Combining equation 9 and equation 10 produces a linear recurrence:

$$\partial^a \partial^b q_m^{ab}(t+1) = \sum_n G_{mn} \langle \phi'^a \phi'^b \rangle_n (\partial^a \partial^b q_n^{ab}(t) + w_n^2 \delta_{t,0}), \qquad A_{mn} \equiv G_{mn} \langle \phi'^a \phi'^b \rangle_n .$$

(11)

Because $q^{ab}$ depends only on earlier inputs, the initial condition is $\partial_\theta^a \partial_\theta^b q_n^{ab}(t) = 0$ for $t \leq 0$. Iterating equation 11 once at $t = 0$ gives

$$\partial_\theta^a \partial_\theta^b q_m^{ab}(1) = \sum_{n=1}^M A_{mn} w_n^2,$$

(12)

Since the network only receives the input at $t = 0$, each successive iteration amounts to matrix multiplication by $A_{mn} \equiv G_{mn} \langle \phi'^a \phi'^b \rangle_n$. Repeating the recurrence $t$ times results:

$$\partial^a \partial^b q_{m,t+1}^{ab} = \sum_n (\mathbf{A}^{t+1})_{mn} w_n^2 .$$

(13)

### A.2.5 CLOSED-FORM FISHER INFORMATION

Finally, substituting Eq. (13) into equation 8 gives

$$\left\langle \left( \frac{\partial \mu_m(t)}{\partial \theta} \right)^2 \right\rangle_J = \sum_n (\mathbf{A}^t)_{mn} w_n^2, \quad A_{mn} \equiv G_{mn} \langle (\phi')^2 \rangle_n .$$

(14)

Eq. equation 14 shows that the propagation of the Fisher information through the network can be effectively captured by the Fisher information diffusion operator $A$.

### A.2.6 ANALYTIC DERIVATION OF THE FISHER INFORMATION DIFFUSION OPERATOR

To obtain an analytical expression for the Fisher information in relation to the optimal connectivity parameters $G_{mn}$ and to gain an intuitive understanding of Fisher information from a network perspective, two key tasks are essential:

1. Analytically resolve the self-consistent equations for $q_1$ and $q_2$ to understand the dynamics in the system. These solutions also allow us to construct Gaussian probability distributions with variances $q_1$ and $q_2$, respectively. From these distributions, we compute the second-order moments $\langle S^2 \rangle$ and $\langle (S')^2 \rangle$, which are essential for determining the Fisher information diffusion operator.

2. Derive an analytical formula for Fisher information that elucidates the relationship between network characteristics, the connection between populations of neurons, and optimal information capacity.

### A.2.7 ANALYTIC CALCULATION OF THE MEAN SQUARED HYPERBOLIC TANGENT

Since both the self consistent equations

$$q_1 = \sigma_1^2 = \sigma^2 + G_{11} \langle S^2 \rangle_1 + G_{12} \langle S^2 \rangle_2 ,$$
$$q_2 = \sigma_2^2 = \sigma^2 + G_{22} \langle S^2 \rangle_2 + G_{21} \langle S^2 \rangle_1 .$$

(15)

and the Fisher information diffusion operator $A_{mn} = G_{mn} \langle (S')^2 \rangle)_n$. With $S = \tanh(x)$, $S^2$ and $S'$ are highly nonlinear and non local, the values are not close to 0 or 1. As a result, using Taylor expansion of the $\tanh(x)$ produces both poor approximation and analytic challenge when calculating the Gaussian average $\langle S^2 \rangle$ and $\langle (S')^2 \rangle$. We notice that both expressions $S^2$ and $S'$ only relate to the some form of Gaussian average $\langle \tanh^2(x) \rangle$, and we can approximate the $\tanh^2(x)$ with $1 - \exp \frac{x^2}{2\epsilon^2}$ (See Fig. S1 ). Note that this expression insures $(1 - \exp \frac{x^2}{2\epsilon^2})|_{x=0} = 0$.The optimal parameter $\epsilon = 0.7784$ can be derived from the minimizing the integral difference $\int_{-\infty}^{\infty} |1 -$

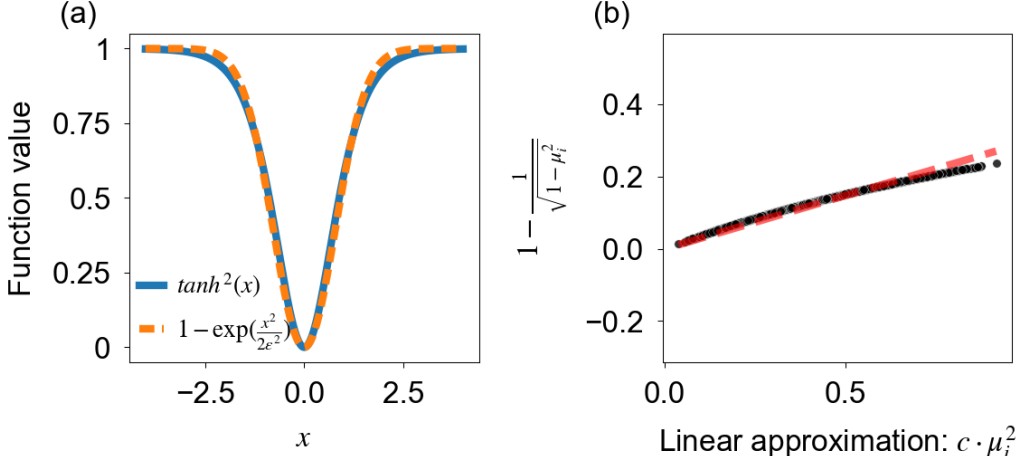

Figure S1: (a) Approximation of the nonlinear function $\tanh^2(x)$ using the surrogate form $f(x, \epsilon) = 1 - \exp(-x^2/2\epsilon^2)$, with the optimal parameter $\epsilon = 0.7784$. (b) Linear approximation of the expression $1 - \frac{1}{\sqrt{1+\mu_i^2}}$ using $c \cdot \mu_i^2$, where the optimal slope $c = 0.2948$ is determined via least-squares fitting. Each point $\mu_i$ is obtained from a grid search over network configurations in the two-population case, constrained to operate near the edge of chaos ($|\rho(A) - 1| < 0.1$), where $\rho(A)$ denotes the spectral radius of the effective connectivity matrix.

$\exp \frac{x^2}{2\epsilon^2} - \tanh^2(x)|$. The Gaussian average $\langle f(x) \rangle_\sigma = \int_{-\infty}^{\infty} f(x) N(0, \sigma^2)$ can be calculated easily with a simple form:

$$\langle \tanh^2(x) \rangle_i = 1 - \frac{1}{\sqrt{1 + (\frac{\sigma_i}{\epsilon})^2}} = 1 - \frac{1}{\sqrt{1 + \mu_i^2}}, \mu_i = \frac{\sigma_i}{\epsilon} ,$$

$$\langle \tanh'(x)^2 \rangle_i = \langle (1 - \tanh^2(x))^2 \rangle_i = \frac{1}{\sqrt{1 + 2\mu_i^2}} .$$

(16)

For numeric solutions, Eq. (equation 15) (equation 20) extends naturally to an arbitrary number of sub-populations, yield a closed system of nonlinear equations for the variance $q_i$. This system can be efficiently solved using the 'fsolve' function from the 'scipy.optimize' package. Once the $q_i$ are obtained, they are substituted into Eq. (equation 20) to evaluate the Fisher-information diffusion operator.

### A.2.8 ANALYTIC CALCULATION OF THE POPULATION SPECIFIC ORDER PARAMETERS

With Eq. (equation 16), we can rewrite the self consistent equations:

$$\mu_1^2 = \mu^2 + M_{11}(1 - \frac{1}{\sqrt{1 + \mu_1^2}}) + M_{12}(1 - \frac{1}{\sqrt{1 + \mu_2^2}}) ,$$

$$\mu_2^2 = \mu^2 + M_{21}(1 - \frac{1}{\sqrt{1 + \mu_1^2}}) + M_{22}(1 - \frac{1}{\sqrt{1 + \mu_2^2}}) ,$$

$$M_{mn} = G_{mn}/\epsilon^2, \ \mu_i^2 = \sigma_i^2/\epsilon^2, \ \mu^2 = \sigma^2/\epsilon^2 .$$

(17)

Solving the Eq. (equation 17) directly is difficult and will lead to unintuitive expression since this is a system of cubic equations with non uniform power in each term. For systems operating near the edge of chaos—characterized by a spectral radius close to one ($|\rho(A) - 1| < 0.1$)—the variable $\mu_i$ remains small (Fig S1 ). In this regime, we can approximate the nonlinear expression $1 - \frac{1}{\sqrt{1+\mu_i^2}}$ using a linearized form. Specifically, we use a least-squares fit to determine the optimal slope $c$ in the following approximation:

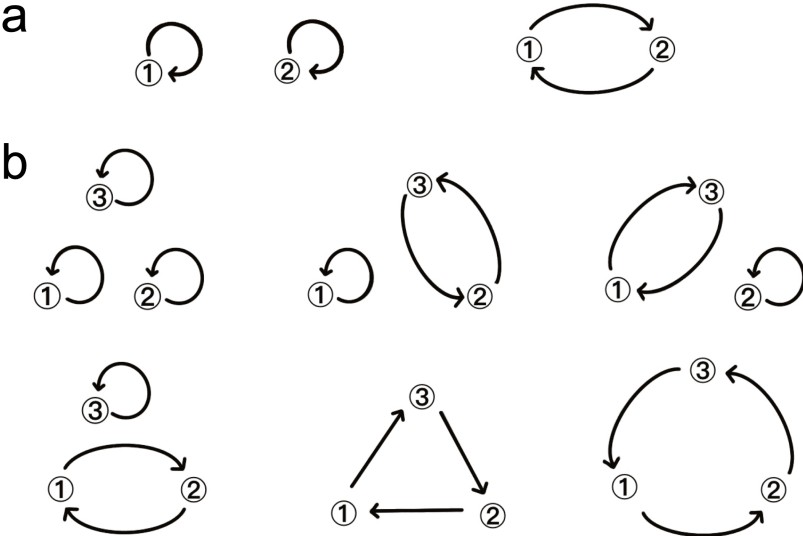

Figure S2: A illustration of the determinant of the weight matrix for 2 populations(a) and 3 populations (b).

$$1 - \frac{1}{\sqrt{1+\mu_i^2}} \approx c\mu_i^2, \quad i \in \{1,2\}, \quad c = 0.2948 . \tag{18}$$

This approximation simplifies further analysis while preserving accuracy in the small-$\mu_i$ limit.

With the linear approximation Eq. (equation 18), the solutions to the self consistent equations are:

$$\mu_1^2 \approx \frac{\mu^2 + c\mu^2(M_{12} - M_{22})}{1 - c\,\mathrm{Tr}(M) + c^2\det(M)} ,$$
$$\mu_2^2 \approx \frac{\mu^2 + c\mu^2(M_{21} - M_{11})}{1 - c\,\mathrm{Tr}(M) + c^2\det(M)} , \tag{19}$$
$$\det(M) = M_{11}M_{22} - M_{12}M_{21},\ \mathrm{Tr}(M) = M_{11} + M_{22} .$$

Under the meanfield, subpopulation 1 and subpopulation 2 are considered as nodes with weight matrix as $M_{mn} = G_{mn}/\epsilon^2$. This mapping makes every term in the analytic expression of Eq. (equation 19) a familiar graph invariant. The trace,$\mathrm{Tr}(M) = \sum_m M_{mm}$, equals the total weight of all self-loops in the network. Meanwhile, the determinant, $\det(M)$, via the Leibniz expansion, becomes a signed sum over all cycle covers (loop configurations), each monomial corresponding to a distinct set of loops weighted by the product of edge weights Harary (1962). These loop configurations are illustrated in Fig. S2.

### A.2.9 ANALYTIC EXPRESSION OF THE FISHER INFORMATION DIFFUSION OPERATOR

The Fisher information diffusion operator $A_{mn} = G_{mn}\langle (S')^2 \rangle_n = \epsilon^2 M_{mn}\langle (S')^2 \rangle_n$. In Eq. (equation 19), we have derived analytic expression in terms of the scaled conductivities $M_{mn} = G_{mn}/\epsilon^2$ and plugging in Eq. (equation 16). Directly substitute the Eq. (equation 19) into Eq. (equation 16), we get:

$$\langle (S')^2 \rangle_1 = \frac{1}{\sqrt{1 + 2\mu_1^2}} \approx f(M)((1 + c(M_{12} - M_{22})) \, ,$$

$$\langle (S')^2 \rangle_2 = \frac{1}{\sqrt{1 + 2\mu_2^2}} \approx f(M)((1 + c(M_{21} - M_{11})) \, ,$$

$$f(M) = \frac{\mu^2}{1 - c \, \mathrm{Tr}(M) + c^2 \det(M)} \, , \tag{20}$$

$$\tilde{f}(M) = 1 - 2c \frac{\mu^2}{1 - c \, \mathrm{Tr}(M) + c^2 \det(M)} = 1 - 2c \, f(M) \, ,$$

$$c = 0.2948 \, .$$

$$A = \epsilon^2 \begin{pmatrix} M_{11} \langle (S')^2 \rangle_1 & M_{12} \langle (S')^2 \rangle_2 \\ M_{21} \langle (S')^2 \rangle_1 & M_{22} \langle (S')^2 \rangle_2 \end{pmatrix} = \epsilon^2 \begin{pmatrix} M_{11} & M_{12} \\ M_{21} & M_{22} \end{pmatrix} \begin{pmatrix} \langle (S')^2 \rangle_1 & 0 \\ 0 & \langle (S')^2 \rangle_2 \end{pmatrix} \, , \tag{21}$$

$$\epsilon = 0.7784 \, .$$

At criticality—i.e. on the "edge of chaos"—the Fisher-information diffusion operator $A$ acquires an eigenvalue exactly equal to unity. Equivalently:

$$\det \big( I - A_{\mathrm{opt}} \big) = 0. \tag{22}$$

By expanding $\det(I - A)$ for our two-population system and grouping terms, we obtain the expression for the condition for the edge of chaos in a fully symmetric form with respect to subpopulations:

$$
\begin{aligned}
0 &= \det(I - A_{opt}) \\
&= 1 - \epsilon^2 (\langle (S')^2 \rangle_1 M_{11} + \langle (S')^2 \rangle_2 M_{22}) + \epsilon^4 \langle (S')^2 \rangle_1 \langle (S')^2 \rangle_2 \det(M) \\
&= 1 - \epsilon^2 \, L_1(M) + \epsilon^4 \, \det(M) \, L_2(M) \, , \\
L_1(M) &= [\tilde{f}(M) - 2c^2 \, f(M) \, \mathrm{Tr_{off}}(M)] \, \mathrm{Tr}(M) + 2c^2 \, f(M)[M_{11}d_2 + M_{22}d_1] \, , \\
L_2(M) &= \tilde{f}^2(M) + 2c^2 f(M)[\mathrm{Tr}(M) - \mathrm{Tr_{off}}(M)] \\
&\quad + 4c^4 f^2(M)(d_1 d_2 - \mathrm{Tr}(M) \, \mathrm{Tr_{off}}(M)) \, , \\
f(M) &= \frac{\mu^2}{1 - c \, \mathrm{Tr}(M) + c^2 \det(M)}, \quad \tilde{f}(M) = 1 - 2c \, f(M) \, , \\
d_i &= \sum_k M_{ik} = M_{i1} + M_{i2}, \quad \mathrm{Tr_{off}}(M) = M_{12} + M_{21}, \, , \\
c &= 0.2948, \; \epsilon = 0.7784 \, .
\end{aligned}
\tag{23}
$$

where we recognize:

1. **Trace**, $\mathrm{Tr}(M) = M_{11} + M_{22}$.
   The *total self-loop weight* (sum of length-1 cycles), which sets first-order feedback gain.
2. **Off-diagonal trace, $\mathrm{Tr_{off}}(M) = M_{12} + M_{21}$.**
   The total cross-population coupling, measuring the strength of two-node interactions.
3. **Determinant,** $\det(M) = M_{11}M_{22} - M_{12}M_{21}$.
   A signed sum over all 2-cycle covers:
   
   - $M_{11}M_{22}$ counts two independent self-loops,
   - $M_{12}M_{21}$ counts the reciprocal 2-node cycle.
   
   The determinant provides insights into the connectivity and spanning trees of a graph, as detailed in the Matrix-Tree Theorem Harary (1962).
4. **Weighted in-degrees.**

$$d_i = \sum_k M_{ik} = M_{i1} + M_{i2},$$

the total incoming weight to subpopulation $i$. The concept of in-degree is a basic measure in graph theory, indicating the number of edges arriving at a node Diestel (2005).

Here, through algebraic manipulation and careful rearrangement, we derive a form of the edge-of-chaos condition that is symmetric across subpopulations and expressed entirely in terms of familiar graph-theoretic quantities—such as trace, off-diagonal trace, determinant, and in-degree of the connectivity matrix. This reformulation reveals how the topology of structured neural networks directly shapes the onset of criticality.

### A.3 Numeric calculation of Fisher information from Monte-Carlo

In the main text, we benchmark the analytic expression of the Fisher information against a direct Monte-Carlo estimate obtained from explicit simulations of the recurrent neural network (RNN). The numerical procedure consists of three main stages: (i) initialization of the random block-structured connectivity, (ii) simulation of neural trajectories under baseline and perturbed inputs, and (iii) estimation of derivatives via symmetric finite differences.

#### A.3.1 Network initialization

The network consists of $N$ neurons partitioned into $M$ subpopulations with sizes $n_m = f_m N$. Synaptic connectivity is represented by a block-structured random matrix $J \in \mathbb{R}^{N \times N}$. Each block $J_{mn}$ is sampled i.i.d. from a Gaussian distribution

$$J_{kl} \sim \mathcal{N}\left(0, \frac{g_{mn}^2}{N}\right), \qquad k \in \text{pop}_m, \ l \in \text{pop}_n,$$

where $g_{mn}$ denotes the population-dependent gain parameter. This choice controls the effective recurrent gain while ensuring that connectivity statistics remain stable as $N$ increases. Here $k$ indexes a postsynaptic neuron in population $m$ and $l$ indexes a presynaptic neuron in population $n$. This blockwise construction ensures that the recurrent connectivity statistics are determined by the gain matrix $g$ while preserving the correct population sizes.

**Dynamical simulation.** Neural activity is described by pre-activations $x_t \in \mathbb{R}^N$ and firing rates $S_t = \tanh(x_t)$. The recurrent dynamics evolve according to

$$x_{t+1} = J S_t + \sigma \xi_t, \qquad \xi_t \sim \mathcal{N}(0, I),$$

with additive Gaussian noise of variance $\sigma^2$. At initialization, an external input $\theta$ is injected into the first population, implemented by setting $x_{1:n_1} \leftarrow \theta$. Multiple trajectories are simulated in parallel to estimate ensemble averages.

#### A.3.2 Perturbation protocol

To estimate the Fisher information with respect to the input parameter $\theta$, we simulate network dynamics under three input conditions: baseline $\theta$, positively perturbed $\theta + \Delta\theta$, and negatively perturbed $\theta - \Delta\theta$. For each condition, we record the full trajectory of neural activities $\{\mathbf{h}(t)\}_{t=1}^T$. Throughout the simulations, we set the baseline input to $\theta = 0$.

#### A.3.3 Fisher information estimation

The sensitivity of mean activity to $\theta$ is approximated via symmetric finite differences:

$$\frac{\partial \mu_t}{\partial \theta} \approx \frac{\mu_t(\theta + \Delta\theta) - \mu_t(\theta - \Delta\theta)}{2\Delta\theta},$$

where $\mu_t(\theta)$ is the average firing rate at time $t$ across trajectories. Squaring and averaging these derivatives over neurons within population $m$ yields a time-resolved Fisher information stored in each subpopulation about the input stimulus over time as in Eq equation 4

Here, we also show the MSE between the simulated fisher information and the analytic prediction of the fisher information in Fig. S3

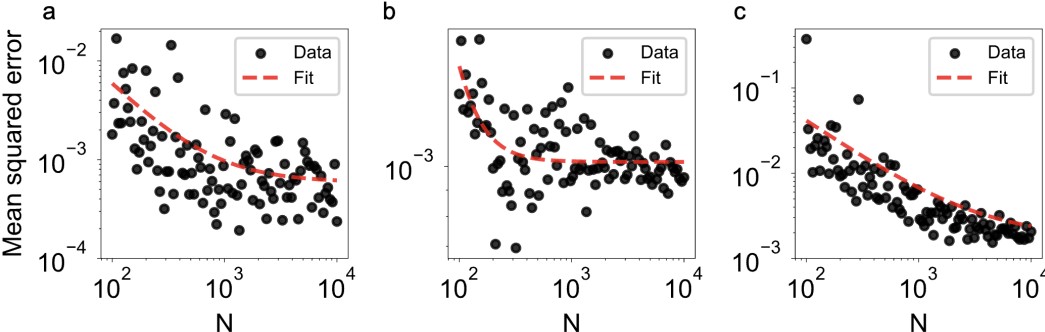

Figure S3: **Convergence of simulated Fisher information to analytic predictions.** Mean-squared error (MSE) between Fisher information trajectories obtained from simulations and from the analytic diffusion operator, across the three connectivity configurations in Fig. 2a–c. The MSE decreases rapidly with network size and becomes negligible for $N \geq 1000$. An exponential fit (dashed line) is shown to highlight the convergence trend.

### A.4 TEST OF FISHER INFORMATION WITH NATURAL IMAGES AS INPUT

**Experimental setup.** To evaluate whether Fisher information predicts geometry preservation, we tested the framework using natural inputs. We used 15,619 IndoorCVPR_09 images, each flattened to a 7,500-dimensional vector, and presented them to the first subpopulation of a two-population recurrent network with $N = 15,000$ neurons, $f_1 = f_2 = 0.5$, and noise variance $\sigma = 0.1$. Each image was processed independently, and at each time step $t$ we recorded the activities of both subpopulations.

**Measuring geometry preservation.** Geometry preservation was quantified by comparing pairwise distances between images in the input space to pairwise distances between their corresponding neural representations. Specifically: 1. For the input set, we computed all pairwise Euclidean distances $D_{\text{input}}(i, j) = \|x_i - x_j\|_2$ between the flattened image vectors. 2. For the network representations at time $t$, we computed analogous pairwise distances $D_{\text{rep}}(i, j) = \|h_i(t) - h_j(t)\|_2$ for each subpopulation. 3. To assess how faithfully the network preserved geometry, we calculated the Pearson correlation coefficient between the upper triangular entries of the two distance matrices,

$$\rho(t) = \text{corr}(\text{vec}(D_{\text{input}}), \ \text{vec}(D_{\text{rep}}(t))),$$

where $\rho = 1$ indicates perfect isometry (exact geometry preservation) and lower values indicate increasing distortion.

This procedure yields a time series $\rho_m(t)$ for each subpopulation $m$, quantifying how input geometry is preserved over time as information diffuses through the network.

**Interpretation.** Although this metric differs from Fisher information, it recovers the same qualitative behavior. In particular, the analytic framework predicts both (i) the oscillatory dynamics of information flow across subpopulations and (ii) the relative ability of different connectivity motifs to preserve the geometry of natural images.

In the main text Theorem 1, we provided an intuitive argument—rooted in compressed sensing and the Restricted Isometry Property (RIP)—that a Fisher-optimal (geometrically neutral) initialization approximately preserves pairwise distances for any sufficiently high-dimensional input ensemble, provided the input sparsity remains below the effective dimensionality of the network (Foucart & Rauhut, 2013). In this regime, the encoding dynamics are expected to generalize across datasets and input statistics.

To test this prediction, we repeated the full geometry-preservation analysis using the CIFAR-10 dataset (32×32 RGB images across 10 classes), scaled to the same dimensionality as the Indoor-CVPR_09 images (flattened to dimension 7500). The resulting information-preservation dynamics are nearly identical to those shown in Fig.2d–f. Quantitatively, the Pearson correlations between the

IndoorCVPR and CIFAR-10 information-flow trajectories are exceptionally high (0.993, 0.992, and 0.980; all $p \ll 10^{-60}$), confirming that the Fisher-optimal initialization induces dataset-independent encoding dynamics.

Our framework views memory as dynamic geometry preservation rather than static attractor storage. This perspective naturally supports generalization at the encoding stage: once initialized at the Fisher-optimal point, the network preserves the relational geometry of novel, unseen inputs without additional training. The consistency of results across IndoorCVPR and CIFAR-10 demonstrates this theoretical prediction—geometry-preserving dynamics arise from the structure of the Fisher-optimized initialization itself, not from dataset-specific learning.

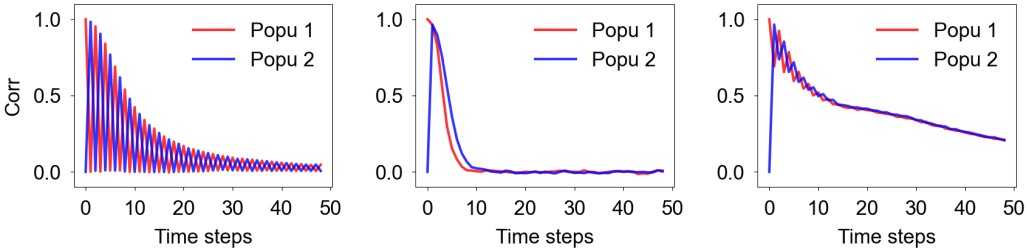

Figure S4: **Geometry preservation for CIFAR-10 inputs.** Analysis analogous to Fig. 2, showing pairwise correlations between input distances and network representations. Architectures predicted to optimize Fisher retention also best preserve input geometry. The Pearson correlations between the IndoorCVPR and CIFAR-10 information-flow trajectories (0.993, 0.992, and 0.980; all $p \ll 10^{-60}$) confirm that Fisher-information–optimized initialization preserves pairwise distances independently of the specific input ensemble, provided the input sparsity is below the effective dimensionality of the network.

### A.5 PROOF OF OPTIMAL STRUCTURE FOR FISHER INFORMATION IN A CHAINED LINEAR NETWORK

In the linear limit—when the activation function is purely linear—one has $\langle (S')^2 \rangle_n = 1$ for all subpopulations. The sensitivity block in the Fisher diffusion operator therefore reduces to the identity, so that

$$A_{\text{linear}} = G.$$

Without loss of generality, consider a chain of four subpopulations with unit self-recurrence:

$$A_{\text{linear}} = G = \begin{pmatrix} 1 & G_{12} & 0 & 0 \\ G_{21} & 1 & G_{23} & 0 \\ 0 & G_{32} & 1 & G_{34} \\ 0 & 0 & G_{43} & 1 \end{pmatrix}. \tag{24}$$

Optimal Fisher information requires that the spectral radius of $A$ equals one, i.e. the largest eigenvalue satisfies $\lambda_{\max} = 1$. Equivalently,

$$0 = \det(A - I) = \det \begin{pmatrix} 0 & G_{12} & 0 & 0 \\ G_{21} & 0 & G_{23} & 0 \\ 0 & G_{32} & 0 & G_{34} \\ 0 & 0 & G_{43} & 0 \end{pmatrix} = G_{12}G_{21}G_{34}G_{43}.$$

For efficient information transmission, the input must enter the first subpopulation and propagate forward through the chain. Hence the forward gains $G_{21}$ and $G_{43}$ cannot vanish. To satisfy $\det(A - I) = 0$, we therefore require

$$G_{12} = 0 \quad \text{or} \quad G_{34} = 0.$$

This condition eliminates the global feedback loops that would otherwise close the chain, giving rise to the broken-loop optimal structure illustrated in Fig. 4. The exact values for the block gain matrix $G$ for each optimized network is shown in Fig. S5

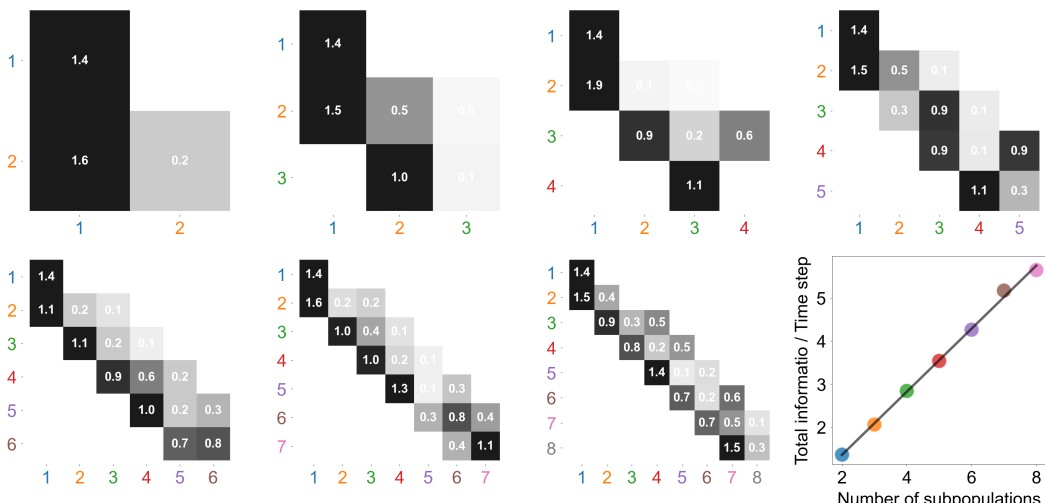

Figure S5: Optimized connectivity matrices $G$ for networks composed of 2 to 8 neuronal subpopulations (insets). Each tick color corresponds to a different number of subpopulations. Matrix element $G_{mn}$ represents the strength of the connection from subpopulation $n$ to subpopulation $m$. The final panel illustrates that the total Fisher information $\overline{\mathcal{I}}$ scales approximately linearly with the number of subpopulations in the optimally connected network.

### A.6 ADDITIONAL EXPERIMENTAL TEST ON SEQUENTIAL MEMORY TASK

### A.6.1 COPY TASK

**Test on multiple delays.** To further validate our theoretical predictions, we conducted additional sequential-memory experiments across multiple delay lengths ($T_{\text{delay}} = 40, 50, 60$) and multiple random seeds (see Figs. S6, S7, and S8). Networks initialized with Fisher–information–optimized weights retain sensitivity to input perturbations occurring far in the past, thereby preserving stimulus information over long times while preventing vanishing or exploding gradients. This leads to more stable dynamics and more efficient training of the decoder.

**Comparison with the common initialization schemes.** We further compared Fisher–optimized initialization with standard schemes, including Xavier, Kaiming, orthogonal, and unitary initialization. Across all delay lengths and seeds, Fisher–information–optimized initialization consistently achieved lower loss, higher accuracy, and faster convergence. These results support the theoretical prediction that operating near the Fisher–optimal regime preserves temporal sensitivity and stabilizes gradient flow, enabling more reliable and efficient learning in sequential-memory tasks.

### A.6.2 APPLYING FISHER–OPTIMAL INITIALIZATION TO LSTM GATES.

We further evaluated whether Fisher–information–optimized initialization can provide benefits in more complex recurrent architectures such as LSTMs and GRUs. For clarity, we recall the standard forward pass of an LSTM with forget gate (Hochreiter & Schmidhuber, 1997):

$$
\begin{aligned}
f_t &= \sigma_g(W_f x_t + U_f h_{t-1} + b_f), \\
i_t &= \sigma_g(W_i x_t + U_i h_{t-1} + b_i), \\
o_t &= \sigma_g(W_o x_t + U_o h_{t-1} + b_o), \\
\tilde{c}_t &= \sigma_c(W_c x_t + U_c h_{t-1} + b_c), \\
c_t &= f_t \odot c_{t-1} + i_t \odot \tilde{c}_t, \\
h_t &= o_t \odot \sigma_h(c_t).
\end{aligned}
\tag{25}
$$

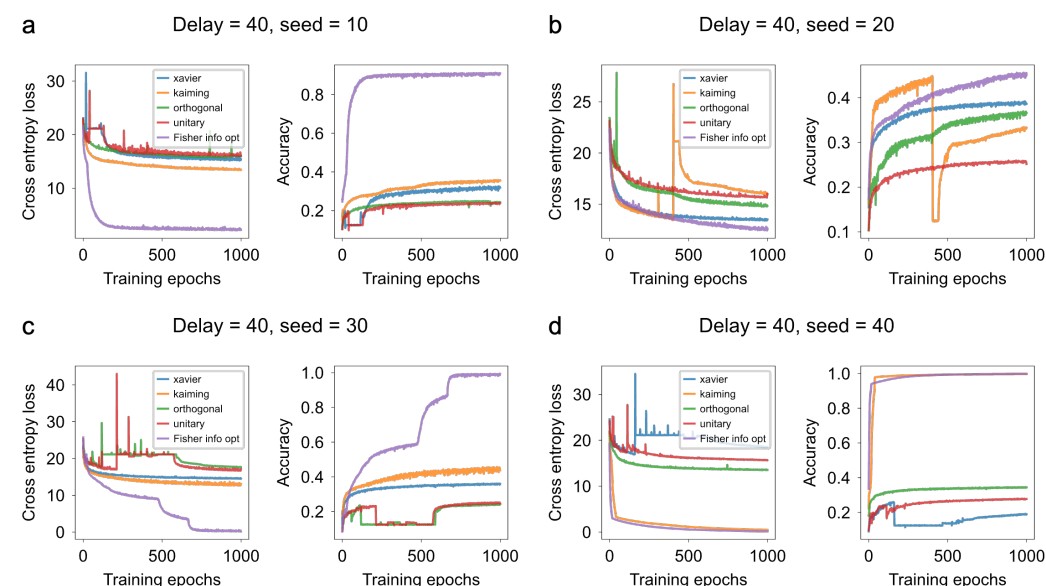

Figure S6: **Copy Task with RNN.** Comparison of Fisher–information–optimized initialization with standard initialization schemes (Xavier, Kaiming, orthogonal, and unitary). Shown are the cross-entropy loss and accuracy during training on a copy task with delay length $T_{\text{delay}} = 40$, evaluated across four independent random seeds.

Unlike the simple RNN model studied in our theory, LSTMs maintain *two* hidden states ($c_t$ and $h_t$), and the dynamics cannot be written as a single recurrent update of the form

$$h_t = \phi(W_{ih}x_t + W_{hh}h_{t-1} + b),\qquad(26)$$

which is the setting under which our Fisher–information analysis is derived. Consequently, our theory does *not* describe the full Fisher-information flow of an LSTM or GRU, and we do not claim any architectural advantages or theoretical optimality for these models.

However, an important observation is that *each gate* in an LSTM computes a recurrent transformation of the form

$$\text{gate}_t = \sigma(Wx_t + Uh_{t-1} + b),\qquad(27)$$

which is structurally identical to the RNN update analyzed in our framework. Thus, while the theory does not extend to the entire LSTM architecture, it is still meaningful to ask whether initializing these gate-level recurrent components using Fisher–optimal weights improves training performance on tasks requiring long-range memory.

To test this, we replaced the RNN in the copy task with an LSTM and initialized the recurrent matrices of each gate using our Fisher–optimal rule, comparing against standard schemes (Xavier, Kaiming, orthogonal, and unitary). As shown in Fig. S9, Fisher–optimized initialization consistently ranks among the top-performing schemes across training runs.

These results provide empirical evidence that—even though our theory is developed strictly for RNNs—the Fisher–optimal initialization of the *RNN-like components* inside an LSTM can still enhance temporal sensitivity, reduce gradient degradation, and improve learning efficiency on long-distance sequential-memory tasks. Importantly, this observation supports the generality of the information-flow perspective, while remaining consistent with the limitations of our theoretical framework.

### A.6.3 SEQUENTIAL MNIST

Following the main text, we include additional experiments testing the training performance of networks initialized with Fisher–information–optimized weights versus standard schemes (Xavier, Kaiming, orthogonal, and unitary). Because the recurrent weights are fixed in this setup, the task

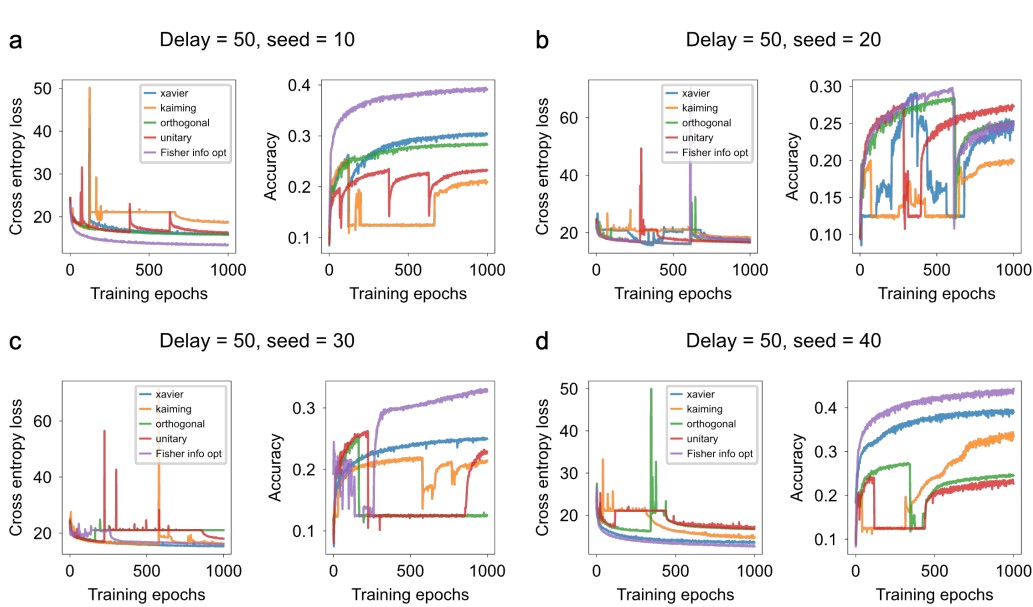

Figure S7: **Copy Task with RNN.** Comparison of Fisher–information–optimized initialization with standard initialization schemes (Xavier, Kaiming, orthogonal, and unitary). Shown are the cross-entropy loss and accuracy during training on a copy task with delay length $T_{\text{delay}} = 50$, evaluated across four independent random seeds.

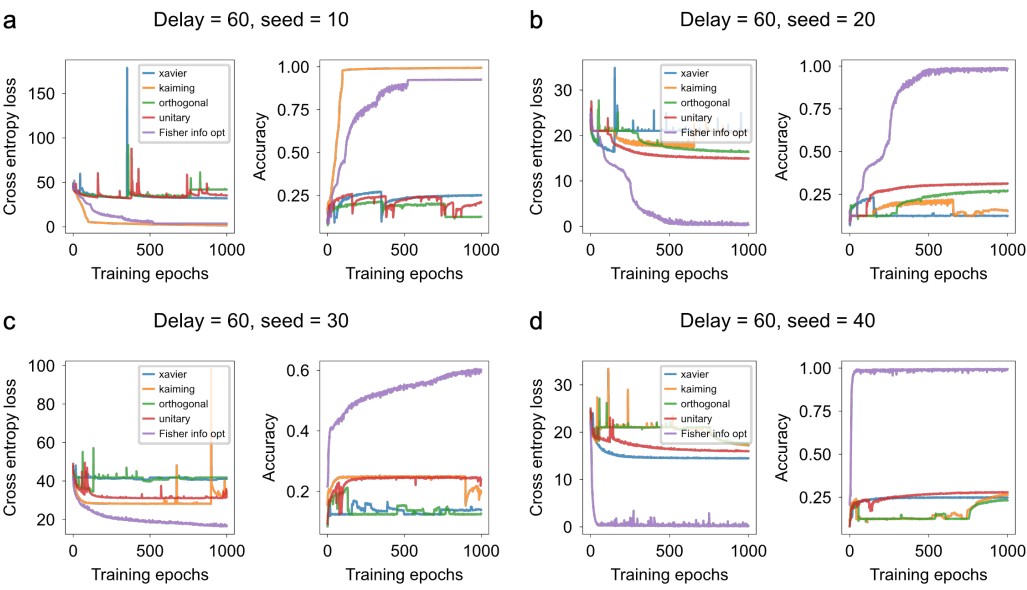

Figure S8: **Copy Task with RNN.** Comparison of Fisher–information–optimized initialization with standard initialization schemes (Xavier, Kaiming, orthogonal, and unitary). Shown are the cross-entropy loss and accuracy during training on a copy task with delay length $T_{\text{delay}} = 60$, evaluated across four independent random seeds.

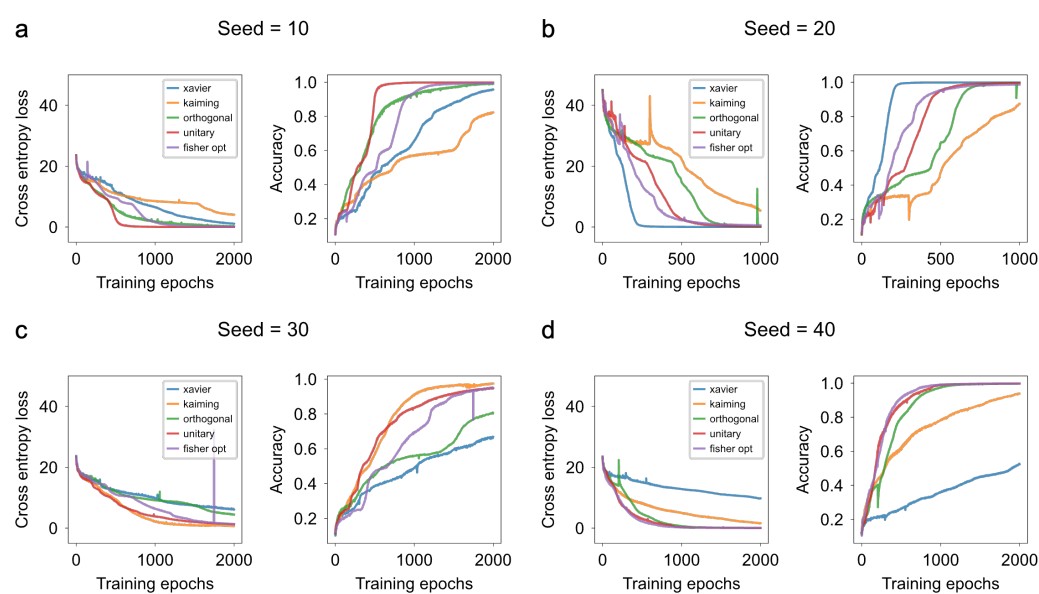

Figure S9: **Copy task with application of Fisher–optimized initialization to LSTM gates.** Each gate of the LSTM is initialized using the Fisher–information–optimized weights derived for recurrent networks. Although our theoretical framework formally applies to single–state RNNs, and thus does not fully capture the dual–state dynamics of LSTMs, the Fisher–based initialization still provides stable gradient propagation in practice. Shown are the cross-entropy loss and accuracy on the copy task with delay length $T_{\text{delay}} = 50$, compared with standard initialization schemes (Xavier, Kaiming, orthogonal, and unitary) across four independent random seeds.

provides a direct assessment of how different initializations affect the network's ability to preserve information from inputs far in the past. In turn, this tests how well the preserved memory supports the effective training of the decoder (Fig. S10).

Across four random seeds, Fisher–information–optimized initialization yields consistently lower loss, higher accuracy, and faster convergence than all baseline initializations. These results confirm that Fisher-derived initializations improve long-range memory by keeping the neural activity at later time steps sensitive to perturbations in early inputs, thereby stabilizing gradient flow and facilitating more effective decoder training.

**The theory is general across input distributions.** Our theoretical results show that Fisher–information–optimal initialization preserves pairwise distances between inputs under the mean-field limit. Because these initialization rules are derived analytically—rather than learned from any specific dataset—the theory predicts that they will preserve input perturbations for *any* stimulus distribution, provided the network is sufficiently large relative to the sparsity of the inputs. This requirement directly parallels the conditions for approximate isometry in compressed-sensing theory and the Johnson–Lindenstrauss lemma (Foucart & Rauhut, 2013), where high-dimensional random projections preserve pairwise distances with high probability.

In Fig. S4, we show that the Fisher-information dynamics across subnetworks remain consistent across two qualitatively different datasets (CIFAR-10 and IndoorCVPR), regardless of whether the network is exactly at the Fisher-optimal point. This demonstrates that the structure of the initialization—and the resulting information flow—is governed by the theoretical framework rather than by dataset-specific statistics. The predicted population-level information dynamics therefore generalize naturally across domains.

To further evaluate this generality in a behavioral task, we conducted sequential-classification experiments on both CIFAR-10 and IndoorCVPR. CIFAR-10 has similar spatial resolution to MNIST ($32 \times 32$ vs. $28 \times 28$) with ten classes. IndoorCVPR is substantially more challenging, containing

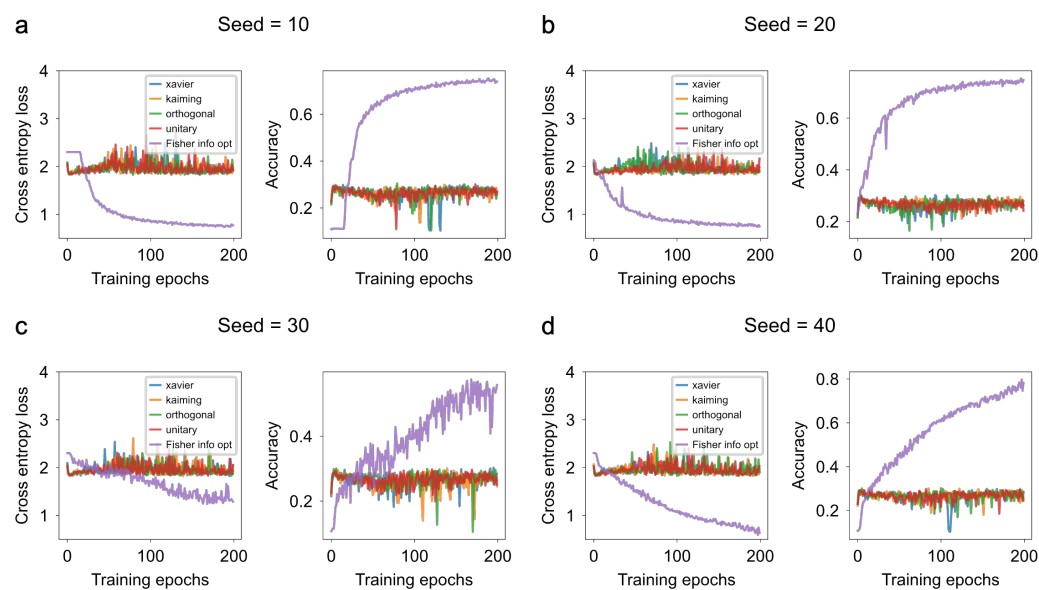

Figure S10: **Sequential MNIST with fixed RNN weights.** Training performance of networks initialized with Fisher–information–optimized weights compared against standard schemes (Xavier, Kaiming, orthogonal, and unitary). Plotted are the cross-entropy loss and accuracy on the Sequential MNIST task, evaluated across four independent random seeds.

high-resolution images ($128 \times 128$) spanning 67 categories. Learning all 67 classes would require a substantially more expressive decoder, which is outside the scope of this work: our focus is on the encoding and memory dynamics of the recurrent network, not on optimizing a deep classification head. Therefore, to maintain the same simple architecture used throughout the paper—an RNN paired with a single-layer MLP decoder—we restricted IndoorCVPR to five representative classes. This setup ensures that differences in performance primarily reflect the network's memory properties rather than decoder complexity.

Across both datasets (Fig. S11 S12), Fisher–information–optimized initialization consistently yields lower loss, higher accuracy, and faster convergence than standard initialization schemes (Xavier, Kaiming, orthogonal, and unitary). These results mirror those obtained on Sequential MNIST and strongly support the theoretical prediction: Fisher-optimal initialization enhances long-range memory by maintaining sensitivity of the recurrent activity at late time steps to small perturbations in early inputs. This stabilizes gradient flow and allows the decoder to train more reliably and efficiently on sequential tasks.

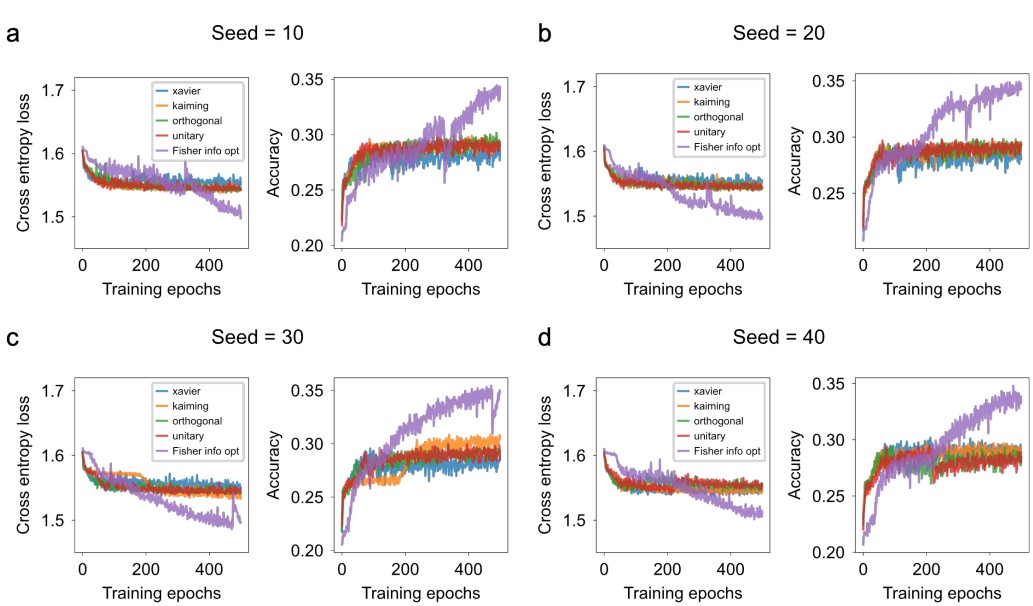

Figure S11: **Sequential classification on CIFAR10 with fixed RNN weights.** Using the same Sequential-MNIST framework, we apply the RNN to the CIFAR10 dataset. Training performance is shown for networks initialized with Fisher–information–optimized weights compared against standard schemes (Xavier, Kaiming, orthogonal, and unitary). Plotted are the cross-entropy loss and accuracy on the sequential classification task, evaluated across four independent random seeds.

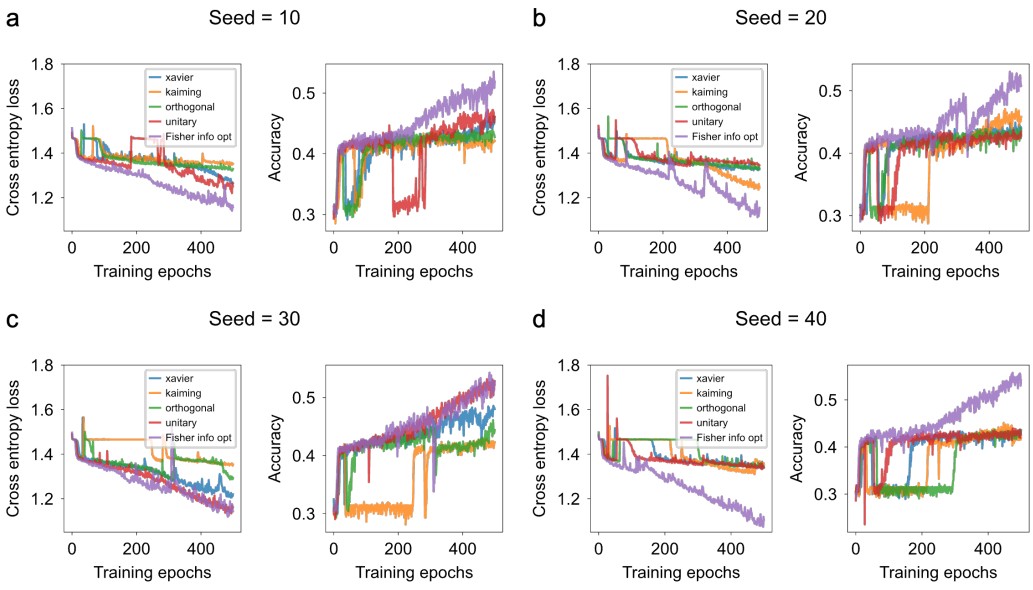

Figure S12: **Sequential classification on IndoorCVPR with fixed RNN weights.** Using the same Sequential-MNIST framework, we apply the RNN to the IndoorCVPR dataset restricted to five classes (airport_inside, artstudio, auditorium, bakery, bar). Training performance is shown for networks initialized with Fisher–information–optimized weights compared against standard schemes (Xavier, Kaiming, orthogonal, and unitary). Plotted are the cross-entropy loss and accuracy on the sequential classification task, evaluated across four independent random seeds.

