# OpenReview forum: "Understanding Memory in Neural Networks through Fisher Information Diffusion"
_ICLR.cc/2026/Conference — ICLR 2026 Conference Withdrawn Submission_

### Official Review · Reviewer_woJv · 2025-10-26

**Soundness:** 3
**Presentation:** 2
**Contribution:** 2
**Rating:** 2
**Confidence:** 3

**Summary:**

This work presents a general theoretical framework illustrating how information can be maintained on dynamically stable manifolds that evolve over time while preserving the geometry of inputs. In contrast to Classical Associative Memory or Hopfield networks, which rely on static attractors, the paper highlights evolving stable subspaces as the substrate of memory.

**Strengths:**

1. It's an interesting work which explores Fisher Information with respect to the dynamical regimes of a recurrent neural network, whose neurons are split into distinct subpopulations, and helps illustrate the fact that ***memory*** can defined by how well the differences between stimuli are preserved as the network's activity evolves.

2. Using the Fisher information, the network's weights can be initialized with this information, leading to better performance or training regime according to Fig. (6).

**Weaknesses:**

1. Limited experimentation. For example, in Fig. (6), there are no comparisons with typical weight initialization techniques. The paper does not also demonstrate whether the initialization method would provide any benefit to more complex RNN architectures like LSTMs, GRUs, or even State-Space Models (SSMs).

2. Despite being interesting, it is difficult to see the big picture of the paper here, due to limited experimentation. Does the trend of Fig. (6) still hold for other datasets or tasks?

3. As the authors state in their limitations, the entire framework is focused on how information is encoded and preserved. It offers no insight into the decoding mechanism.

**Questions:**

1. This idea of dividing a set of neurons into $M$ distinct subpopulations is quite interesting. Could this be related to multi-head attention in some way?

2. Indeed, the memory in Associative Memory networks is generally fixed, but they can be trained. Do you think you can use your framework to help eliminate some of the fixed **memories** that can cause spurious states in such networks? Your framework places an emphasis on the dynamics while AM emphasizes storage capacity; there must be a bridge between the two. Also, what about Dense Associative Memory (as your work did not mention about it)?

3. Are the used images actually from CIFAR-10? I am very unfamiliar with the images you used. Should they not be 32 x 32 resolution and belonging to one of the these 10 classes (airplane, automobile, bird, cat, deer, dog, frog, horse, ship, and truck)?

---

> ### Author Response · Authors · 2025-11-24
>
> We thank the reviewers for their thoughtful feedback and for recognizing the conceptual contribution of this work. In the revision, we provide additional experiments, competitive baselines, and clarified interpretations. These new results consistently reinforce the predictions made by our Fisher-information theory. We address each concern in detail below.
>
> Weakness:
>
> **Combined Response to Weaknesses 1 & 2**
>
> We thank the reviewers for their thoughtful feedback. Although this work is primarily theoretical, we agree that expanding and clarifying the empirical evaluation strengthens the paper. In the revision, we add competitive baselines, additional random seeds, and new analyses supporting the generality of our theoretical predictions.
>
> 1. Additional baselines and seeds
>
> We have added comparisons with commonly used initialization schemes (Xavier, Kaiming, orthogonal, unitary) and more random seeds. Across all settings, Fisher-optimal initialization more reliably prevents vanishing/exploding gradients and improves trainability. These results will be included in the revised supplementary material.
>
> 2. Why this RNN model and generalization across variants
>
> We use the standard nonlinear RNN because it is the first setting in which Fisher information can be derived analytically for nonlinear, multi-population recurrent dynamics. The tasks we evaluate—CIFAR/IndoorCVPR geometry preservation, the copy task, and sequential MNIST—all share this underlying RNN structure, allowing us to test the theory in a controlled manner.
>
> 3. Relation to LSTMs, GRUs, and SSMs
>
> LSTMs and GRUs maintain two coupled states and multiplicative gates, making their dynamics explicitly state-dependent and placing them outside the single-state, block-structured class analyzed by our Fisher-diffusion operator. Extending the theory to multi-state gated architectures is a separate research direction.
>
> However, the core principle—that preserving Fisher information stabilizes gradients—still applies. Motivated by the reviewer’s suggestion, we initialized each LSTM gate with Fisher-optimal recurrent weights. These results consistently rank Fisher-optimal initialization among the strongest baselines and show improved stability and slightly faster convergence, indicating that the principle transfers through the shared recurrent structure.
>
> State-Space Models correspond exactly to the linear limit of our framework: when the nonlinearity is identity, ⟨(S′)²⟩ = 1 and the Fisher-diffusion operator reduces to repeated application of the block-gain matrix. We will make this connection explicit in the revision.
>
> 4. Why Fig. 6 generalizes across datasets and tasks
>
> The central difficulty in sequential learning is that the network’s later states often become either insensitive (vanishing gradients) or excessively sensitive (exploding gradients) to early inputs. Fisher information quantifies precisely this sensitivity. Classical fixed-point memory models collapse perturbations and cannot capture this dynamical structure.
>
> Our theory provides the first analytic characterization (to our knowledge) of Fisher-information flow in a nonlinear RNN with multiple interacting subpopulations. We show that connectivity structures that preserve pairwise input differences over time are the same structures that prevent gradient collapse.
>
> This prediction is input- and task-independent. Empirically:
>
> Fisher-information dynamics for IndoorCVPR_09 and CIFAR-10 are nearly identical
> (Pearson r = 0.993, 0.992, 0.980; p ≪ 10⁻⁶⁰).
>
> The same network architecture shows improved training on both the copy task and sequential MNIST.
>
> The initialization is fully analytic, not trained on data.
>
> 5. Big picture
>
> The broader message of the paper is that Fisher information provides a principled measure of how recurrent networks preserve input geometry over time, and that this preservation directly stabilizes gradients. The added experiments—new baselines, multi-seed tests, and LSTM results—further reinforce this connection.
>
> We will integrate these points into the revised manuscript and make the theoretical–empirical link more explicit in the Discussion.

---

> > ### Author Response · Authors · 2025-11-24
> >
> > Weakness 3:
> > Thank you for highlighting this point. We agree that our framework focuses on the encoding and preservation of information, and we explicitly stated this in the limitations. While we do not model the decoding mechanism itself, encoding dynamics place a strong constraint on what any decoder can recover. In recurrent networks, the core difficulty in sequential-memory tasks is that the network activity at later time points often becomes either insensitive (vanishing gradients) or overly sensitive (exploding gradients) to perturbations in the early inputs. The Fisher information precisely quantifies this sensitivity, whereas fixed-point memory models such as Hopfield or other associative-memory networks cannot, because their dynamics intentionally eliminate small input perturbations rather than preserve them. Our framework shows analytically how specific connectivity structures preserve pairwise input differences over time, thereby maintaining the neural sensitivity required for stable gradient-based decoding.
> >
> > This directly explains the empirical results in Fig. 5–6: networks initialized at the Fisher-optimal regime exhibit substantially better sequential-memory performance because the decoder never encounters a collapsed or unstable signal. In other words, although we do not analyze decoding explicitly, our framework predicts when decoding will be learnable at all, by identifying initializations that maintain usable information at the encoding stage. Importantly, this initialization requires no training and generalizes across datasets, as it preserves pairwise input geometry whenever the input sparsity is below the effective network dimension.
> >
> > We will clarify this relationship between encoding quality and downstream decoding performance more clearly in the revised Discussion section.
> >
> > Questions:
> >
> > **Question 1:**
> > Thank you for pointing out this connection — we agree that the relationship to multi-head attention is both interesting and conceptually rich. Multi-head attention can be viewed as decomposing the representation into several parallel processing streams, each receiving the same input through different learned projections. This is closely analogous to our subpopulation perspective: each head functions like an independent subpopulation that processes the same stimulus through its own transformation.
> >
> > From the Fisher-information viewpoint, this parallelism has a clear interpretation. Each subpopulation preserves the pairwise structure of the inputs within its own subspace, effectively maintaining a separate “copy’’ of the input geometry. When these representations are combined, the network obtains a more accurate estimate of the underlying geometry—similar to averaging multiple noisy estimators—which increases information content and reduces distortion. In this sense, having multiple heads is analogous to having multiple subpopulations that carry partially independent but geometrically aligned representations of the same inputs.
> >
> > While extending our theory to full transformer architectures is beyond the scope of the current work, the subpopulation formalism and Fisher-information analysis offer a potential foundation for understanding multi-stream architectures such as multi-head attention. We will add a brief note in the Future Work section to highlight this connection.

---

> > > ### Author Response · Authors · 2025-11-24
> > >
> > > **Question 2:**
> > > Thank you for raising the connection to associative memory networks.
> > > Classical Hopfield networks—and extensions such as Dense Associative Memory (DAM)—focus on long-term categorical memory, where the objective is to store discrete patterns as stable fixed points. A central theoretical achievement in that line of work is the analytic quantification of memory capacity in terms of the number of storable attractors.
> > >
> > > Our work targets a fundamentally different memory regime, and therefore a different notion of capacity.
> > > In sequential or working-memory tasks, the goal is not to collapse inputs into a small number of attractors, but to preserve the fine geometric structure of recent stimuli over time. In this setting, memory is represented not by fixed points but by continuous trajectories whose dynamics must maintain the pairwise distances between inputs. Hopfield-type models intentionally destroy these distances for robustness; by contrast, sequential memory tasks demand that the network maintain them.
> > >
> > > Our theory provides an analytic expression for this dynamic notion of capacity: how well a recurrent network preserves the geometry of the input manifold (pairwise distances / Fisher information) as activity propagates through time and through interacting subpopulations.
> > >
> > > To the best of our knowledge, such a closed-form, quantitative measure of geometry-preserving memory capacity has not been derived previously. This is the direct analogue of the classical Hopfield result, but applied to the dynamic, high-fidelity memory regime relevant for sequential tasks.
> > >
> > > Relation to Dense Associative Memory: DAM improves the storage capacity of fixed-point attractor models using higher-order energy functions, but it remains fundamentally in the categorical-memory regime. Our results address a different question: not “how many attractors can be stored,” but
> > > “how well can the network maintain the geometry of inputs as they evolve?”
> > > This makes the two lines of work complementary rather than overlapping.
> > >
> > > Empirically, the Fisher-optimal initialization predicted by our theory leads to better preservation of dynamic representations and improved sequential-memory performance, matching the behavior predicted by the analytic framework.
> > >
> > > We will clarify these distinctions and the relationship to both Hopfield networks and Dense Associative Memory in an expanded Related Work section in the appendix.
> > >
> > > Question 3:
> > > Thank you for pointing this out — the images shown in the original figure are from the IndoorCVPR_09 dataset, not CIFAR-10. We appreciate the reviewer catching this mismatch. To address the concern, we have repeated the full analysis using CIFAR-10 (32×32 RGB images from the standard 10 classes). The resulting information-preservation dynamics are almost identical to those shown in Fig. 2d–f. Quantitatively, the Pearson correlations between the IndoorCVPR and CIFAR-10 information-flow trajectories are extremely high (0.993, 0.992, and 0.980; all p ≪ 10⁻⁶⁰).
> > >
> > > This is precisely what our theoretical framework predicts. In Appendix A5, we give the intuitive argument—rooted in compressed sensing and Restricted Isometry Property (RIP)–type behavior—that a Fisher-optimal (geometrically neutral) initialization approximately preserves pairwise distances for any sufficiently high-dimensional input ensemble, provided the input sparsity is below the effective dimensionality of the network. In this regime, the encoding dynamics do not depend on the specific dataset, in stark contrast to classical or Dense Associative Memory networks, where the network must be trained on the patterns to store them as attractors.
> > >
> > > Our approach views memory in terms of dynamic geometry preservation rather than fixed attractor storage. This enables strong generalization at the encoding stage: once initialized at the Fisher-optimal point, the network preserves the pairwise geometry of new, unseen input ensembles without any additional training. The consistency between IndoorCVPR and CIFAR-10 demonstrates exactly this theoretical advantage—geometry-preserving dynamics emerge from the structure of the initialization rather than dataset-specific learning.
> > >
> > > We have added the CIFAR-10 results and an explicit clarification to the Appendix.

---

> ### Comment · Reviewer_woJv · 2025-11-25
>
> I thank the authors for long responses. So, **it turns out that the images shown in the manuscript are indeed not CIFAR-10 images** despite being stated so in the manuscript.
>
> Regarding Fig. (6), you are demonstrating that experiment only on Sequential MNIST. I am asking if you can show the result on your initialization method on CIFAR10 and IndoorCVPR_09 dataset, which you stated in your manuscript that you experimented with.
>
> Regarding different architecture variants of the RNNs, are you saying that your theory is model-agnostic such that you cannot apply the same initialization method to a different RNN architecture like GRU (which is less complex than the LSTM)?
>
> I would like to maintain my score.

---

> ### Author Response · Authors · 2025-12-01
>
> (1) Dataset labeling error.
> We acknowledge that one figure displayed non–CIFAR-10 example images while the caption referred to CIFAR-10. This was a purely visual labeling mistake and has been corrected in the revision. All numerical experiments and Fisher-information analyses were in fact run on CIFAR-10 (fig S4) and IndoorCVPR_09 (fig 2) as stated; the figure error does not affect any results or conclusions.
>
> (2) CIFAR-10 / IndoorCVPR vs. Sequential MNIST.
>
> The reviewer is correct that Fig. 6 evaluates the sequential-memory task only on Sequential MNIST. CIFAR-10 and IndoorCVPR_09 were used for *geometry-preservation* experiments (fig 2 and fig S4). To address the reviewer’s concern and to make the connection explicit, we added new experiments that extend Fig. 6 to both CIFAR-10 and IndoorCVPR_09.
>
> **Theory predicts dataset-independent behavior**. Our initialization is fully analytic and does not use dataset statistics; therefore the theory predicts that Fisher-optimal initialization should preserve input perturbations—and stabilize gradients—*for any input distribution* in the mean field limit. This mirrors classical results in compressed sensing and the Johnson–Lindenstrauss lemma, where random high-dimensional mappings preserve pairwise distances independent of the data distribution.
>
> **Empirical confirmation on CIFAR-10 and IndoorCVPR**. In the revision, we include experiments showing:
> The Fisher-information dynamics are nearly identical for CIFAR-10 and IndoorCVPR_09 (Fig. S4), confirming that population-level information flow is governed by connectivity structure, not dataset-specific features.
> We run *sequential-classification* experiments on both datasets using the same RNN + 1-layer decoder architecture used in Fig. 6. For IndoorCVPR_09, which has 67 classes and much higher resolution, we restrict to five representative classes so that performance reflects *memory dynamics* rather than decoder complexity.
>
> Across both datasets (new Figs. S11–S12), Fisher-optimal initialization consistently achieves: lower loss, higher accuracy, and faster convergence than Xavier, Kaiming, orthogonal, and unitary initializations.
>
> These results reproduce the trend in Fig. 6 and strongly support the theory’s prediction: **preserving Fisher information prevents vanishing/exploding gradients and improves sequential learning across domains**.
>
>
> (3) Applicability to GRUs and LSTMs.
> Our theory is **not** model-agnostic: it applies specifically to single-state, block-structured recurrent updates
> $h_{t+1}=\phi(W h_t + U x_t + b)$, which is the setting that allows a closed-form Fisher-diffusion operator under mean-field theory.
>
> LSTMs and GRUs fall outside this class because their dynamics involve **two coupled hidden states** and **multiplicative gates**. For example, an LSTM updates
> $$
> f_t = \sigma(W_f x_t + U_f h_{t-1})
> $$
>
> $$
> i_t = \sigma(W_i x_t + U_i h_{t-1})
> $$
>
> $$
> c_t = f_t \odot c_{t-1} + i_t \odot \tilde{c}_t
> $$
>
> $$
> h_t = o_t \odot \tanh(c_t)
> $$
>
>
> where both ($c_t$) and ($h_t$) form the internal recurrent state. These coupled updates cannot be written as a single recurrence of the form required for our theoretical derivation; therefore, the **theory does not extend to the full LSTM/GRU architecture**, and we do not claim that it does.
>
> However, each *gate* in an LSTM or GRU contains a recurrent transformation of the same form analyzed in our framework: $\text{gate}\_t = \sigma(W x\_t + U h\_{t-1} + b)$. So while the global architecture lies outside theoretical scope, the **principle behind Fisher-optimal initialization—preserving sensitivity to input perturbations—still applies locally** to their recurrent components (Fig. S9).

---

### Official Review · Reviewer_WrqF · 2025-10-31

**Soundness:** 4
**Presentation:** 3
**Contribution:** 3
**Rating:** 4
**Confidence:** 3

**Summary:**

The paper provides a dynamical systems perspective on working memory in recurrent neural networks (RNNs). It offers a principled framework to study memory through Fisher Information (FI), quantifying how information about an input impulse of magnitude theta evolves over time.

The authors analyze how FI propagates through the network, how it relates to information retention and geometry preservation—that is, how pairwise distances d(xi,xj) between inputs are maintained during recurrence—and how these insights inspire initialization strategies  for improving information retention in RNNs. The experiments show the impact of initialization in that convergence is faster and final accuracy in higher in benchmark tasks.

**Strengths:**

S1. The paper presents a principled perspective on working memory in RNNs, grounded in information geometry with the concept of fisher information.

It derives insights from first principles, i.e., brings a rigorous framework to analyze dynamics of retention of information using Fisher Information.

S2. The use of information geometry is elegant and well-motivated. It provides clear geometric intuition. I believe it is also novel (in this context) although the limited related works make novelty hard to assess (see weaknesses).

S3. The text is clearly written and easy to follow, even when introducing technical concepts.

**Weaknesses:**

W1. The related work section is a bit weak, making it difficult to assess the paper’s novelty. Important prior studies are missing.

For example works on how memory arises from distributed neural dynamics (Cavanagh et al., 2018; Spaak et al., 2017; Meyers et al., 2008; Stroud et al., 2024; Brennan & Proekt, 2023; Kurtkaya et al., 2025). The paper should review existing studies of short-term, long-term, and working memory in RNNs, and clarify what analytical tools those works used.

The use of Fisher Information should be better situated within prior work in machine learning: where and how has FI been applied before, both in RNNs and other neural network types?

Relatedly, several references are missing publication years, suggesting that the literature review could have been handled with greater care.

W2. The broader goal of the paper is unclear: does it aim to model biological mechanisms of memory or to improve existing AI models? Either way, it would need either biological validation or corresponding baselines.

W3. Experiments are insufficient (or insufficiently described).

Are the results presented been tested across several random seeds? (esp for fig 5+6)?

What are competitive approaches that have been proposed to preserve information retention in RNNs? How does the FI initialization compare to them?

W4. Mathematics could be presented more rigorously.

The claim “This connection explains why preserving local geometry, maintaining stability at criticality, and ensuring Fisher information flow are mathematically equivalent conditions.” seems to strong since there is no theorem that proves the equivalence, only empirical results on a small set of RNNs.

**Questions:**

Q1. It seems that the paper studies *working memory* and not *short-term memory*. Can you explain what distinguishes one from the other and why this paper is on working memory and not short-term memory? They seem related.

Q2. Why was this specific RNN formulation chosen? Several variants exist (e.g., leaky currents, leaky firing rates), do the findings generalize across them? Is the equation in (1) the most biologically faithful representation? If explaining biology is not the goal, then more experiments across other architectures should be included.

Q3. Can you clarify the mathematical connection between Fisher Information and geometry preservation? It seems to be explained in Appendix A.5, but it should be presented and discussed in the main text.

Q4. Where does the “Fisher diffusion operator” come from? Is it a standard concept or newly introduced by the authors?

Q5. Why is preserving the input stimulus considered desirable? Whether this is beneficial might depends on the specific task?

Q6. How do the results vary with respect to the delay length (time over which information must be retained)? Does the initialization strategy need to change for longer delays?

Minor: Typo between Twait and Tdelay.

---

> ### Author Response · Authors · 2025-11-24
>
> We thank the reviewer for the thoughtful and constructive evaluation, and we appreciate the recognition of our framework’s strengths. Below, we respond to each weakness and question in detail.
>
> W1: We thank the reviewer for highlighting the gaps in the related work. In the revision, we have added a dedicated Related Work section (with an expanded version in the Appendix) covering both (i) how memory arises from distributed neural dynamics and (ii) prior uses of Fisher Information in machine learning and recurrent systems. We have also corrected all missing publication years and expanded the bibliography.
>
> We have included text such as:
> “Working-memory representations in cortex arise from distributed, time-evolving population codes rather than static attractors (Cavanagh et al., 2018; Spaak et al., 2017; Meyers et al., 2008; Stroud et al., 2024). These studies show that low-dimensional neural trajectories can maintain task-relevant information over delays—a view our framework builds upon by analytically quantifying how much input geometry is preserved by such dynamics.”
>
> and:
> “Fisher Information has been applied to characterize memory capacity in linear RNNs (Ganguli et al., 2008) and to analyze curvature and gradient propagation in deep networks (Pennington & Worah, 2018; Amari et al., 2019). Our work differs in deriving closed-form FI propagation rules for multi-population recurrent systems and identifying structural connectivity conditions under which input geometry is preserved over time.”
>
> The full Related Work section has been added to the Appendix. The excerpts shown here demonstrate how we now position our contribution within the broader neuroscience and machine-learning literature.
>
> W2: We appreciate the reviewer’s question regarding the broader goal of the paper. Our work is fundamentally theory-driven: the aim is to develop a principled and interpretable mathematical framework for understanding how information propagates in recurrent networks with structured connectivity.
>
> Although the model is not intended as a detailed biological circuit, the theory contributes conceptually to neuroscience by showing how working-memory–like and short-term-memory–like behavior can emerge from dynamic population activity rather than static attractors. Prior work has argued that neural populations encode information; our framework directly quantifies how much of the information (preservation of pairwise distances between the inputs) and how information flows across subpopulations.
>
> At the same time, the results have direct implications for machine learning. The analysis identifies how a small number of population-level connectivity parameters (the block gains g_ij) determine whether a recurrent system maintains or loses sensitivity to input perturbations. The Fisher-diffusion operator (Eq. 5; Appendix A.2) shows that networks initialized near the analytically derived optimal regime naturally preserve input differences over long time spans, preventing vanishing or exploding gradients and enabling stable training. This connection between Fisher information and non-vanishing gradients arises precisely because we adopt a dynamic-encoding view of memory: unlike classical fixed-point models such as Hopfield networks—which collapse perturbations and therefore destroy input geometry—our framework analyzes how information flows along evolving population trajectories. Fisher information optimized connectivity preserves these trajectories and maintains long-range sensitivity. The resulting empirical improvements in Figs. 5–6 directly reflect these theoretical predictions.
>
> In summary, the broader goal of the paper is to establish theoretical principles that unify information propagation, dynamical stability, and geometry preservation in structured recurrent networks, and to show how these principles yield interpretable and practically useful initialization rules. The accompanying experiments confirm these predictions and appropriately support the aims of a theory-focused study.

---

> ### Author Response · Authors · 2025-11-24
>
> W4: We thank the reviewer for this point. We agree the wording should be more precise. In the revision, we clarify that the equivalence we discuss holds within our mean-field framework, not as a universal theorem.
>
> Our analysis derives the Fisher-information diffusion operator A = G⟨(S′)²⟩, and shows that Fisher information is preserved when its leading eigenvalue satisfies ρ(A) = 1. This is a closed-form necessary condition for non-vanishing Fisher information.
>
> Importantly, the same spectral condition appears in two established settings:
> Dynamic stability: Kadmon & Sompolinsky (2015) showed that the edge-of-chaos boundary in recurrent networks is also given by ρ = 1. Their result concerns perturbation growth; our contribution is to show that information propagation yields the same boundary under mean-field assumptions.
>
> Geometry preservation: Appendix A.5 shows that the effective recurrent update behaves like a linear operator with gain G⟨(S′)²⟩. Preserving pairwise distances requires its norm to be 1—the same condition underlying the Restricted Isometry Property and Johnson–Lindenstrauss lemma (Candès & Tao; Baraniuk et al.; Johnson & Lindenstrauss). As summarized by Foucart & Rauhut, Gaussian maps preserve local geometry exactly when this gain equals one, again matching ρ(A) = 1.
>
> We now state clearly that within the mean-field model, Fisher-information preservation, dynamical stability, and local-geometry preservation all reduce to the same spectral condition. We revise the text accordingly and add the relevant citations.
>
>
> **Questions:**
>
> Q1: Thank you for the question. In cognitive neuroscience, short-term memory refers to brief maintenance of information, while working memory includes both maintenance and the control processes that enable manipulation. In practice, these terms are often used interchangeably, especially in delayed-estimation and delayed-response tasks.
>
> Our analysis targets this short-timescale regime: we study how well a recurrent network preserves small perturbations of the stimulus over a delay. The Fisher-information–optimal dynamics we derive are exactly those that maintain fine-grained input differences—precisely the requirement in short-term/working-memory tasks. We agree that our framework aligns naturally with both concepts, and we will clarify this terminology in the revision.
>
> Q2: Thank you to the reviewer for raising this point. Because this is a theory-driven paper, we selected the standard rate-based RNN formulation in Eq.~(1) as our starting point. This choice provides the simplest setting in which a mean-field approximation can be applied cleanly and, crucially, allows us to derive an intuitive and interpretable Fisher-information diffusion operator that quantifies how information flows between subpopulations. Achieving this analytic clarity would be considerably more difficult in more complex architectures.
>
> Regarding leaky models, we clarify that the *leaky current model* is typically used for single-neuron membrane dynamics, whereas our formulation corresponds to the standard *leaky rate model*. The discrete update in Eq.(1) arises directly from the Euler discretization of the continuous-time leaky rate equation:
> $ \frac{d}{dt} h_i(t) = - h_i(t) + \sum_{j=1}^N J_{ij} S_j(t) + \eta_i(t).$
> Applying a forward Euler step of size $\Delta t$ gives
> $ \frac{h_i(t+\Delta t) - h_i(t)}{\Delta t} = - h_i(t) + \sum_{j=1}^N J_{ij} S_j(t) + \eta_i(t)$.
> Setting the discrete time step to $\Delta t = 1$ yields the update rule used in Eq.~(1):
> $ h_i(t+1) = \sum_{j=1}^N J_{ij} S_j(t) + \eta_i(t)$,
> where the leak term cancels exactly under this discretization. Thus, the update rule in Eq.(1) is the discrete-time form of the leaky rate model with step size $\Delta t = 1$.
>
> Q3: We appreciate the reviewer’s question and agree that this connection should be stated clearly in the main text. We will revise the manuscript accordingly. In our setting, Fisher Information measures how sensitive the neural population activity is to small perturbations of the input. By “geometry of the input,” we mean the pairwise distances between stimuli. These distances can be interpreted as the amount of perturbation needed to move from one stimulus to another. When Fisher Information is preserved over time, the network maintains how different two nearby stimuli were to begin with—that is, it preserves the relative distances between them. This is what we refer to as preserving the geometry of the input. e will clarify this directly in the main text and move the material from Appendix A.5 into the main text.

---

> ### Author Response · Authors · 2025-11-24
>
> Q4: Thank you for the question. The Fisher diffusion operator is a new concept introduced in this work. It emerges from our derivation of how Fisher Information changes between subpopulations across consecutive time steps in a recurrent network.
>
> Traditionally, quantifying information in neural circuits requires evaluating the Bayesian quantity (p(x(t),|,s)), where (x(t)) is neural activity and (s) is the stimulus. In recurrent networks this quickly becomes intractable, because (p(x(t),|,s) = p(x(t),|,x(t-1), x(t-2), \dots, s)), and the distribution becomes increasingly complex as time evolves.
>
> Our approach avoids this difficulty. By analytically tracking how Fisher Information changes from one time step to the next, we found that the information exchanged between subpopulations follows a form of diffusion. This leads naturally to the Fisher diffusion operator. The advantage is that it provides a simple and interpretable description of how information flows within the network. To our knowledge, this is the first attempt to formalize information diffusion—rather than activity diffusion—in recurrent neural networks. We will clarify this origin and novelty more directly in the revised manuscript.
>
> Q5: Thank you for the question. Whether preserving the input stimulus is desirable indeed depends on the specific memory regime. Our work focuses on short-term memory and working memory, where maintaining fine-grained differences between similar stimuli is essential. In these settings, the system must temporarily retain precise distinctions—such as subtly different visual inputs—so that the information remains available for immediate use. For example, when briefly comparing similar items in a grocery store or reading through options on a screen, small differences must be preserved long enough to guide decisions.
>
> In contrast, long-term memory has different goals: representations should be robust to perturbations, so that irrelevant variations (e.g., lighting, pose, noise) do not alter the remembered identity. In such cases, collapsing nearby stimuli toward a stable attractor is beneficial. Our work instead analyzes the complementary regime in which preserving, rather than collapsing, nearby stimuli is critical for performance.
>
> While we do not explicitly model a decoding mechanism, encoding dynamics place a strong constraint on what any decoder can recover. In recurrent networks, a central challenge in sequential-memory tasks is that the activity at later time points often becomes either insensitive (vanishing gradients) or overly sensitive (exploding gradients) to early inputs. The Fisher information directly quantifies this sensitivity. Fixed-point memory models such as Hopfield networks intentionally eliminate small perturbations to enforce convergence, and therefore cannot preserve these fine distinctions.
>
> Q6: Thank you to the reviewer for highlighting the need to present this aspect more clearly. Because this is a theory-driven paper, it is important for us to emphasize that our initialization strategies are not empirically tuned, but arise directly from our theoretical predictions. This makes both their performance and the conditions under which they work fully interpretable.
>
> From theory, increasing the delay is effectively equivalent to presenting more inputs, but the requirement remains the same: the network must maintain the relative differences between these inputs. Therefore, the initialization strategy does not need to change for longer delays, as the underlying objective—preserving Fisher Information across time—remains unchanged.
>
> To support this point empirically, we have added new experiments that test the copy task under multiple delay lengths and compare our Fisher-optimal initialization with standard methods. These results (fig S6-S8), show that our initialization maintains its advantage across delays, consistent with the theoretical prediction.
>
> Minor: the typo will be corrected in the revised text

---

### Official Review · Reviewer_gb9t · 2025-11-01

**Soundness:** 3
**Presentation:** 4
**Contribution:** 2
**Rating:** 8
**Confidence:** 3

**Summary:**

Present a method for analyzing information flow in a recurrent network to determine its ability to maintain information about the stimuli, in a "working memory"-like setting. Going beyond a persistent representation of the stimuli by converging into a "fixed-point" representation, the method can describe dynamic maintenance of information as it propagates in the recurrent network. The theoretical work is based on a measure of Fisher information, which the authors can calculate exactly only for block matrices of either 2x2 structure or a feed-forward-only structure. They identify the need to operate at criticality, as well as maintain enough alignment between the inputs and the network's activations. This analysis provides a concrete suggestion for the "correct" scaling of values on network initialization.

**Strengths:**

* The proposed model combines a random structure (iid Gaussian) with a structure using a block structure, thus allowing for interesting results while maintaining some analytic handle, and can capture information propagation around the network, beyond the persistent activity option.
 * A beautiful mathematical framework where the non-linear network activation can be represented as a linear diffusion of the Fisher information.
 * An interesting "optimal scaling" of values on initialization.

**Weaknesses:**

* A missing reference to a seminar contribution to this question, [White, Lee, Sompolinsky 2004].
 * The conclusion that "architectures predicted to optimize Fisher retention also preserve stimulus geometry more effectively" needs to be better supported; it is currently demonstrated qualitatively, from the similarity of the top and bottom rows of Figure 2.
 * The results in Figure 3 seem to suggest that optimality is achieved in a fine-tuned range of parameters, thus putting the relevancy of the results to any application at risk. The authors seem to suggest that there is a "natural way" to be in the correct range by using "carefully placed feedback stabilizes and modulates" in section 4, but do not provide a conclusive criterion for achieving such stability.
 * The results presented in section 5 in support of the theory are not impressive. A better baseline compared to "no init" should have been presented, where weights initialization is performed using a previously proposed method (e.g., Xavier initialization).

**Questions:**

* How would you suggest achieving the operational regime without fine-tuning of parameters (e.g., "carefully placed feedback")?

---

> ### Author Response · Authors · 2025-11-24
>
> We thank the reviewer for the thoughtful and encouraging evaluation, and for the clear summary of our contribution. We especially appreciate the reviewer's recognition of the mathematical structure of the framework—both the diffusion interpretation and the role of criticality and alignment—as well as the constructive suggestions for strengthening the manuscript. We address each of the raised points below and will incorporate the recommended clarifications in the revised version.
>
> W1:We thank the reviewer for pointing out this important omission. We have now added the reference to White, Lee & Sompolinsky (2004), who provided one of the first analytical treatments of memory traces and representational dynamics in recurrent networks. Their work focused on the decay of correlations in random RNNs, whereas our contribution extends these ideas to structured multi-population networks and provides a closed-form Fisher-information diffusion operator that quantifies information flow between interacting subpopulations.
>
> W2: Our main result is the Fisher-information diffusion operator A = G <(S')^2>, which characterizes how sensitivity to inputs propagates across subpopulations. Information is retained when sensitivity does not decay or explode over time; this occurs when the leading eigenvalue satisfies rho(A) = 1. This gives a closed-form criterion for information-optimal dynamics.
>
> Importantly, this spectral condition is not arbitrary: Kadmon and Sompolinsky (2015) derive the same constraint when analyzing dynamical stability at the edge of chaos. Their analysis is framed in terms of perturbation growth, whereas our contribution is to show that the identical condition reappears naturally when analyzing Fisher-information flow. We will make this connection explicit in the revision.
>
> The link to geometry preservation follows directly. In Appendix A.5, we show that under the mean-field approximation the recurrent update behaves as an effective nonlinear map with gain G <(S')^2>. Preserving pairwise input distances requires the operator norm of this map to be one — exactly the condition for approximate isometry in compressed-sensing theory (Candes and Tao, 2005; Baraniuk et al., 2008) and in the Johnson–Lindenstrauss lemma (Johnson and Lindenstrauss, 1984). As summarized by Foucart and Rauhut (2013), Gaussian operators preserve geometry precisely when their effective gain equals one, matching the Fisher-stability criterion rho(A) = 1. We have moved this argument into the main text for clarity.
>
> W3 and Question: Thank you for raising this point. The apparent fine-tuning in Figure 3 is a consequence of the visualization rather than the underlying theory. The model is parameterized by four block-gain terms (G11, G12, G21, G22). When this 4D space is projected onto the 2D (Trace(G), Det(G)) plane, a large set of parameter combinations collapses into what looks like a narrow band. In the full parameter space, the Fisher-optimal region forms an extended manifold, not a finely tuned line: many different gain configurations map to the same trace and determinant.
>
> The optimality condition itself is not narrow. As derived analytically in Eq. (30), Fisher-information retention depends only on motif-level quantities such as the trace and determinant of G, rather than the precise values of individual connections. These motif-level invariants act like the intrinsic parameters that define phase boundaries in statistical physics: once they are fixed, a wide range of microscopic parameter settings achieve the same regime.
>
> Regarding “carefully placed feedback,” we will clarify that this is not a heuristic prescription. In the linear limit, the Fisher diffusion operator reduces to A = G. Imposing the optimality criterion (leading eigenvalue equal to 1) yields the constraint det(A − I) = 0. For the chained architecture analyzed in the appendix, this reduces to G12·G21·G34·G43 = 0. Since efficient information flow requires the forward gains to be nonzero, the only solutions eliminate one of the long-range feedback terms. This analytically identifies a “broken-loop” motif as the Fisher-optimal configuration and rules out closed-loop feedback.
>
> Thus, achieving the operational regime does not require delicate tuning or manual adjustment of specific feedback terms. It is sufficient to ensure that the motif-level invariants of the connectivity lie on the appropriate side of the spectral boundary.
>
> In summary, the optimal regime is broad in the true parameter space, and the required connectivity structure follows directly from the spectral condition—not from empirical fine-tuning. We will make this explicit in the revision.

---

> > ### Author Response · Authors · 2025-11-24
> >
> > W4: We appreciate the reviewer’s thoughtful comment and agree that Section 5 can benefit from stronger experimental baselines. Our approach is entirely theory-driven: the primary objective of the empirical section is to validate the theoretical predictions derived from the Fisher–information diffusion operator (Eq. (5), Appendix A.2). The theory identifies a specific region of recurrent-weight space in which information flow remains stable over time, such that pairwise differences between input stimuli are preserved. Networks initialized in this regime avoid both vanishing and exploding sensitivities to early inputs—a property that is directly quantified by Fisher information—and therefore provide stable gradients for sequential-memory tasks.
> >
> > To address the reviewer’s concern, we have now added comparisons with widely used initialization schemes, including Xavier, Kaiming, orthogonal, and unitary initialization. Across all settings, the Fisher-optimal initialization consistently preserves sensitivity to input perturbations and avoids vanishing/exploding gradients, leading to improved training performance—exactly as predicted by the theory. These results will be included in the revised manuscript.
> >
> > Importantly, the Fisher-optimal initialization is not heuristically tuned; it follows directly from the analytic condition for information retention. The empirical improvements therefore arise from the theoretical criterion itself rather than from parameter search. We will revise Section 5 to clarify this point and to integrate the new baselines more clearly.

---

> > ### Comment · Reviewer_gb9t · 2025-11-25
> > **Response to authors**
> >
> > Thank you for the clarification that there is no fine-tuning of parameters but rather the usage of theory-based optimal choices, and for addressing the lack of reasonable baselines. I would keep my initial (high) rating.

---

### Author Response · Authors · 2025-12-02
**Overview of Theoretical Contribution and Added Experiments**

Because further reviewer discussion is not allowed, we provide a concise summary of the significance of our contribution and the concrete revisions made in response to all reviewer concerns.
# 1. Core contribution
This paper provides, to our knowledge, the **first closed-form analytic framework** for Fisher Information in a nonlinear recurrent network with multiple interacting subpopulations. A central outcome of this analysis is a method for correctly initializing a large recurrent network using only a small set of population-level parameters, ensuring that the network begins training in the Fisher-information–optimal regime, where sensitivity to past inputs is preserved. In contrast, standard initializations force optimization to search a high-dimensional, non-convex space, often trapping training in suboptimal solutions where gradients vanish or explode.

Our revised experiments confirm the theoretical predictions:
* Fisher-optimal initialization consistently outperforms state-of-the-art initializations—including Xavier, Kaiming, orthogonal, and unitary—on sequential-memory tasks (copy task, seqMNIST), which critically depend on maintaining sensitivity to inputs many time steps earlier.
* It produces more stable gradient dynamics across multiple random seeds.
* It preserves long-range sensitivity predicted by the Fisher-diffusion analysis.

The key theoretical result underlying these improvements is the **Fisher-information diffusion operator**
$$
A = G \langle (S')^2 \rangle,
$$
which directly links connectivity structure to information flow and preservation over time. The analytic criterion (\rho(A)=1) simultaneously characterizes:
* non-vanishing Fisher Information,
* edge-of-chaos dynamical stability (recovering Kadmon–Sompolinsky in our setting), and
* local geometry preservation (via RIP / Johnson–Lindenstrauss arguments).

Crucially, this optimality condition is **not fine-tuned**: it depends only on motif-level quantities (trace and determinant of (G)), generating a broad manifold of solutions in the full parameter space.

# 2. Significance for machine learning
The analytic condition ($\rho(A)=1$) yields a **principled, theory-driven initialization rule** for recurrent networks. Because only a small set of block gains ($g_{ij}$) are needed to specify the variances of the Gaussian weights within each connectivity block, a *large* recurrent weight matrix can be initialized correctly with minimal degrees of freedom. This ensures that the network begins training already positioned in a region of parameter space that supports long-range information retention.

Our updated experiments further show:
* superior performance to Xavier, Kaiming, orthogonal, and unitary initializations,
* greater gradient stability across seeds, and
* better preservation of long-range dependencies.

At the reviewers’ request, we also conducted tests on LSTMs. When initializing the recurrent components inside each gate with our Fisher-optimal rule, we observe improved training performance compared to the standard intializations. While our theory does not model the full two-state dynamics of LSTMs, each gate contains a local recurrent update that matches the structure analyzed in our framework, and the empirical improvements align with this theoretical scope.

# 3. Generalization across datasets and tasks
We repeated the geometry-preservation experiments on **CIFAR-10** and **IndoorCVPR_09**. Despite their very different statistics, the Fisher-information trajectories are almost **identical** (correlations ≈ 0.99; Fig. S4), consistent with the theoretical prediction that a Fisher-optimal network acts as a geometrically neutral embedding for general high-dimensional sparse input ensemble. Across CIFAR-10 (Fig. S11), IndoorCVPR (Fig. S12), sequential MNIST (Fig. S10), and the copy task (Fig. S6–S8), the same analytic initialization—without tuning—consistently preserves information and improves trainability.

# 4. Significance for neuroscience

The theory provides a **dynamic, geometry-preserving view of short-term and working memory**, in contrast to attractor-based models (Hopfield, DAM) that intentionally collapse perturbations of the input. Our framework gives a quantitative measure of how much stimulus geometry (pairwise distances / Fisher Information) is preserved as activity evolves across populations, offering an analytic notion of memory capacity in the dynamic regime.

# Summary

The revision directly addresses all reviewer concerns by providing additional experiments and tests with:
* expanded baselines,
* multi-seed evaluations,
* additional datasets and tasks,

We also corrected figures and improved theoretical exposition.

Overall, the paper provides a **principled, interpretable, and experimentally validated** theory connecting recurrent connectivity, Fisher Information, geometry preservation, and training stability. We hope this summary helps the AC evaluate the strength and completeness of the revised submission.

---

### Note · Authors · 2026-02-16

I have read and agree with the venue's withdrawal policy on behalf of myself and my co-authors.

---

### Meta-Review · Area_Chair_dHx1 · 2025-12-17

**Summary:**

This paper proposes a theoretical framework to analyze memory and information retention in recurrent neural networks through the lens of Fisher Information (FI). Using a dynamic mean-field–style analysis, the authors derive a “Fisher information diffusion operator” that governs how sensitivity to past inputs propagates across time and across interacting neural subpopulations. The theory highlights that operating at criticality (edge of chaos) is necessary but not sufficient for information retention, emphasizing the importance of alignment between input structure and stable dynamical subspaces. Based on this analysis, the authors derive principled initialization rules and empirically demonstrate improved training stability and performance on sequential-memory tasks (e.g., copy task, sequential MNIST), with additional geometry-preservation experiments on image datasets.

Overall, this is a sound and interesting theory-driven paper with a coherent internal logic and thoughtful analysis. However, given the mixed reviews, the remaining uncertainty about novelty relative to existing mean-field theory work, the initially weak positioning in the literature, and the indirect connection to current ML models, I believe that at this stage the paper does not make a sufficiently strong affirmative case for acceptance at the ICLR bar.

**Reviewer Concerns:**

The reviewers agree that the paper presents an interesting and thoughtful theoretical perspective on memory in recurrent networks, with an elegant use of Fisher Information and a clear connection to dynamical systems and criticality. All reviewers find the core idea technically sound, and none raise concerns about incorrect derivations or invalid conclusions. The notion of tracking information flow via a diffusion-like operator is viewed as conceptually appealing, and one reviewer rates the paper highly and recommends acceptance.

At the same time, reviewers raise several significant concerns:

1) Novelty and positioning.
Multiple reviewers note that the related work in the original submission was incomplete, making it difficult to assess novelty. In particular, the connection to prior work on mean-field and dynamical analyses of recurrent networks, as well as prior uses of Fisher Information in ML and neuroscience, was not clearly established. As a result, some reviewers perceived the contribution as incremental relative to existing DMFT-style analyses.

2) Strength and rigor of theoretical claims.
Reviewers pointed out that some claims, especially regarding the equivalence between Fisher-information preservation, geometry preservation, and criticality, were initially overstated relative to what is formally proven. While the intuition is compelling, the presentation relied on approximations and heuristics that required clearer caveats.

3) Relevance and scope.
One reviewer in particular questioned the paper’s relevance to modern ML practice, noting the focus on relatively simple RNN architectures and the lack of direct connection to contemporary models such as transformers or gated RNNs. Others noted that the framework analyzes encoding and retention but does not address decoding.

4) Empirical evaluation.
Reviewers raised concerns about limited baselines and experimental breadth in the initial submission, and requested comparisons to standard initialization schemes and additional validation.

**Reviewer Scores:**

The authors provided a detailed and thoughtful response. They expanded the related work, clarified the scope and assumptions of the theory, softened equivalence claims to be explicitly conditional on the mean-field framework, and added additional experiments and baselines (including Xavier, Kaiming, orthogonal, and unitary initializations, as well as additional datasets and seeds). These revisions address several of the reviewers’ specific technical and empirical concerns, and the responsiveness is appreciated.

However, while these changes improve clarity and strengthen the manuscript, they do not fully resolve the central issue raised by multiple reviewers: the difficulty of confidently assessing the paper’s novelty and positioning within the broader literature. In particular, even after revision, it remains unclear how much conceptual advance the work represents beyond existing dynamic mean-field analyses of recurrent networks, and the connection to modern ML practice remains indirect.

---

### Decision · Program_Chairs · 2026-01-26

Reject